# A Theory of PAC Learnability under Transformation Invariances

**Han Shao**
Toyota Technological Institute Chicago
Chicago, 60637
`han@ttic.edu`

**Omar Montasser**
Toyota Technological Institute Chicago
Chicago, 60637
`omar@ttic.edu`

**Avrim Blum**
Toyota Technological Institute Chicago
Chicago, 60637
`avrim@ttic.edu`

## Abstract

Transformation invariances are present in many real-world problems. For example, image classification is usually invariant to rotation and color transformation: a rotated car in a different color is still identified as a car. Data augmentation, which adds the transformed data into the training set and trains a model on the augmented data, is one commonly used technique to build these invariances into the learning process. However, it is unclear how data augmentation performs theoretically and what the optimal algorithm is in presence of transformation invariances. In this paper, we study PAC learnability under transformation invariances in three settings according to different levels of realizability: (i) A hypothesis fits the augmented data; (ii) A hypothesis fits only the original data and the transformed data lying in the support of the data distribution; (iii) Agnostic case. One interesting observation is that distinguishing between the original data and the transformed data is necessary to achieve optimal accuracy in setting (ii) and (iii), which implies that any algorithm not differentiating between the original and transformed data (including data augmentation) is not optimal. Furthermore, this type of algorithms can even "harm" the accuracy. In setting (i), although it is unnecessary to distinguish between the two data sets, data augmentation still does not perform optimally. Due to such a difference, we propose two combinatorial measures characterizing the optimal sample complexity in setting (i) and (ii)(iii) and provide the optimal algorithms.

## 1 Introduction

Transformation invariances are present in many real-world learning problems. That is, given a certain set of transformations, the label of an instance is preserved under any transformation from the set. Image classification is often invariant to rotation/flip/color translation. Syntax parsing is invariant to exchange of noun phrases in a sentence. Such invariances are often built into the learning process by two ways. One is designing new architectures in neural networks to learn a transformation invariant feature, which is usually task-specific and challenging. A more universally applicable and easier way is data augmentation (DA)[1], that is, adding the transformed data into the training set and training a model with the augmented data. Although DA performs well empirically, it is unclear whether and when DA "helps". In this paper, we focus on answering two questions:

---

[1]Throughout the paper, we refer to ERM over the augmented data by DA.

36th Conference on Neural Information Processing Systems (NeurIPS 2022).

*How does data augmentation perform theoretically?*
*What is the optimal algorithm in terms of sample complexity under transformation invariances?*

We formalize the problem of binary classification under transformation invariances in the PAC model. Given instance space $\mathcal{X}$, label space $\mathcal{Y} = \{0, 1\}$, and hypothesis class $\mathcal{H}$, we consider the following three settings according to different levels of realizability.

(i) Invariantly realizable setting: There exists a hypothesis $h^* \in \mathcal{H}$ such that $h^*$ can correctly classify not only the natural data (drawn from the data distribution) but also the transformed data. For example, considering the transformation of rotating images where all natural images are upright, the hypothesis $h^*$ can correctly classify every upright image (natural data) and their rotations (transformed data).

(ii) Relaxed realizable setting: There exists a hypothesis $h^* \in \mathcal{H}$ such $h^*$ has zero error over the support of the data distribution (and therefore will correctly classify the transformed data that lies in the support of the data distribution), but $h^*$ may not correctly classify transformed data that lies outside the support of the natural data distribution. For example, there exists an $h^*$ classifying all small rotations that lie in the support of the distribution correctly, but misclassifying upside-down cars.

(iii) Agnostic setting: Every hypothesis in $\mathcal{H}$ might not fit the natural data.

In most of this work, we consider the case where the set of transformations forms a group (e.g., all rotations and all color translations), which is a classic setting studied in literature (e.g., Cohen and Welling, 2016; Bloem-Reddy and Teh, 2020; Chen et al., 2020). Some algorithms and analyses in this work also apply to non-group transformations (e.g., croppings).

**Main contributions** First, we show that DA outperforms vanilla ERM but is sub-optimal in setting (i) above. We then introduce a complexity measure (see Definition 4) that characterizes the optimal sample complexity of learning in setting (i), and we give an optimal (up to log-factors) algorithm in this setting based on 1-inclusion-graph predictors. Second, we characterize the complexity of learning in setting (ii) when the learner only receives the augmented data (without specifying which are natural). Such a characterization provides us with a sufficient condition under which DA "hurts". Third, we introduce a complexity measure (see Definition 5) that characterizes the optimal sample complexity of learning in settings (ii) and (iii) above, and we give optimal algorithms for these settings. Finally, we also provide adaptive learning algorithms that interpolate between settings (i) and (ii), i.e., when $h^*$ is partially invariant. We want to emphasize that our complexity measures take into account the complexity of both the hypothesis class $\mathcal{H}$ and the set of transformations being considered. The results are formally summarized in Section 3.

**Related work** Theoretical guarantees of DA has received a lot of attention recently. Chen et al. (2020); Lyle et al. (2020) study theoretical guarantees of DA under the assumption of "equality" in distribution, i.e., for any transformation in the transformation group, the data distribution of the transformed data is approximately the same as that of the natural data (e.g., the upside-down variations of images happen at the same probability as the original upright images). Under this assumption, they show that DA reduces variance and induces better generalization error upper bounds. Our work does not make such an assumption. Dao et al. (2019) models augmentation as a Markov process and shows that for kernel linear classifiers, DA can be approximated by first-order feature averaging and second-order variance regularization components. The concurrent work by Shen et al. (2022) studies the benefit of DA when training a two layer convolutional neural network in a specific multi-view model, showing that DA can alter the relative importance of various features. There is a line of theoretical study on the invariance gain in different models. For example, Elesedy and Zaidi (2021) study the linear model and Elesedy (2021); Mei et al. (2021); Bietti et al. (2021) study the non-parametric regression. The concurrent work by Elesedy (2022) also studies PAC learning under transformation invariances but only provides an upper bound on the sample complexity, while our work provides a complete characterization of learning under this model with optimal algorithms. There is a parallel line of theoretical study on architecture design (e.g., Wood and Shawe-Taylor, 1996; Ravanbakhsh et al., 2017; Kondor and Trivedi, 2018; Bloem-Reddy and Teh, 2020).

Learning under transformation invariances has also been studied a lot empirically. Here we briefly mention a few results. DA has been applied as standard method in modern deep learning, e.g., in Alexnet (Krizhevsky et al., 2012). Gontijo-Lopes et al. (2020) proposes two measures, affinity and diversity, to quantify the performance of the existing DA methods. Fawzi et al. (2016); Cubuk

et al. (2018); Chatzipantazis et al. (2021) study how to automatically search for improved data augmentation policies. For architecture design, one celebrated example is convolutions (Fukushima and Miyake, 1982; LeCun et al., 1989), which are translation equivariant. See Cohen and Welling (2016); Dieleman et al. (2016); Worrall et al. (2017) for more different architectures invariant or equivariant to different symmetries.

Another line of related work is adversarial training, which adds the perturbed data into the training set and can be considered as a special type of data augmentation. Raghunathan et al. (2019); Schmidt et al. (2018); Nakkiran (2019) study the standard accuracy of adversarial training and provide examples showing that adversarial training can sometimes "harm" standard accuracy.

**Notation** For any $n \in \mathbb{N}$, let $\mathbf{e}_1, \mathbf{e}_2, \ldots$ denote the standard basis vectors in $\mathbb{R}^n$. For any set $\mathcal{V}$ and any $\mathbf{v} \in \mathcal{V}^n$, let $\mathbf{v}_{-i} = (v_1, \ldots, v_{i-1}, v_{i+1}, \ldots, v_n) \in \mathcal{V}^{n-1}$ denote the remaining part of $\mathbf{v}$ after removing the $i$-th entry and $(v', \mathbf{v}_{-i}) = (v_1, \ldots, v_{i-1}, v', v_{i+1}, \ldots, v_n) \in \mathcal{V}^n$ denote the vector after replacing $i$-th entry of $\mathbf{v}$ with $v' \in \mathcal{V}$. Let $\oplus$ denote the bitwise XOR operator. For any $h \in \mathcal{Y}^{\mathcal{X}}$ and $X = \{x_1, \ldots, x_n\} \subset \mathcal{X}$, denote $h_{|X} = (h(x_1), \ldots, h(x_n))$ the restriction of $h$ on $X$. A data set or a sample is a multiset of $\mathcal{X} \times \mathcal{Y}$. For any sample $S$, let $S_{\mathcal{X}} = \{x | (x, y) \in S\}$ (with multiplicity) and for any distribution $\mathcal{D}$ over $\mathcal{X} \times \mathcal{Y}$, for $(x, y) \sim \mathcal{D}$, let $\mathcal{D}_{\mathcal{X}}$ denote the marginal distribution of $x$. For any data distribution $\mathcal{D}$ and any hypothesis $h$, the expected error $\mathrm{err}_{\mathcal{D}}(h) := \Pr_{(x,y) \sim \mathcal{D}}(h(x) \neq y)$. Denote $\mathrm{err}(h) = \mathrm{err}_{\mathcal{D}}(h)$ when $\mathcal{D}$ is clear from the context. For any sample $S$ of finite size, $\mathrm{err}_S(h) := \frac{1}{|S|} \sum_{(x,y) \in S} \mathbb{1}[h(x) \neq y]$. For any sample $S$ of possibly of infinite size, we say $\mathrm{err}_S(h) = 0$ if $h(x) = y$ for all $(x, y) \in S$.

## 2  Problem setup

We study binary classification under transformation invariances. We denote by $\mathcal{X}$ the instance space, $\mathcal{Y} = \{0, 1\}$ the label space and $\mathcal{H}$ the hypothesis class.

**Group transformations** We consider a group $\mathcal{G}$ of transformations acting on the instance space through a mapping $\alpha : \mathcal{G} \times \mathcal{X} \mapsto \mathcal{X}$, which is compatible with the group operation. For convenience, we write $\alpha(g, x) = gx$ for $g \in \mathcal{G}$ and $x \in \mathcal{X}$. For example, consider $\mathcal{G} = \{e, g_1, g_2, g_3\}$ where $e$ is the identify function and $g_i$ is rotation by $90i$ degrees. Given an image $x$, $ex = x$ is the original image and $g_1 x$ is the image rotated by 90 degrees. The orbit of any $x \in \mathcal{X}$ is the subset of $\mathcal{X}$ that can be obtained by acting an element in $\mathcal{G}$ on $x$, $\mathcal{G}x := \{gx | g \in \mathcal{G}\}$. Note that since $\mathcal{G}$ is a group, for any $x' \in \mathcal{G}x$, we have $\mathcal{G}x' = \mathcal{G}x$. Thus we can divide the instance space $\mathcal{X}$ into a collection of separated orbits, which does not depend on the data distribution. Given a (natural) data set $S \subset \mathcal{X} \times \mathcal{Y}$, we call $\mathcal{G}S := \{(gx, y) | (x, y) \in S, g \in \mathcal{G}\}$ the augmented data set.

**Transformation invariant hypotheses and distributions** To model transformation invariance, we assume that the true labels are invariant over the orbits of natural data. Formally, for any transformation group $\mathcal{G}$ and $X \subset \mathcal{X}$, we say a hypothesis $h$ is $(\mathcal{G}, X)$-invariant if

$$h(gx) = h(x), \forall g \in \mathcal{G}, x \in X.$$

That is to say, for every $x \in X$, $h$ predicts every instance in the orbit of $x$ the same as $x$. For any marginal distribution $\mathcal{D}_{\mathcal{X}}$ over $\mathcal{X}$, we say a hypothesis $h$ is $(\mathcal{G}, \mathcal{D}_{\mathcal{X}})$-invariant if $h(gx) = h(x)$ for all $g \in \mathcal{G}$, for all $x \in \mathrm{supp}(\mathcal{D}_{\mathcal{X}})$, i.e., $\Pr_{x \sim \mathcal{D}_{\mathcal{X}}}(\exists x' \in \mathcal{G}x : h(x') \neq h(x)) = 0$. We say a distribution $\mathcal{D}$ over $\mathcal{X} \times \mathcal{Y}$ is $\mathcal{G}$-invariant if there exists a $(\mathcal{G}, \mathcal{D}_{\mathcal{X}})$-invariant hypothesis $f^*$ (possibly not in $\mathcal{H}$) with $\mathrm{err}_{\mathcal{D}}(f^*) = 0$. We assume that the data distribution is $\mathcal{G}$-invariant throughout the paper.

**Realizability of hypothesis class** We consider three settings according to the different levels of realizability of $\mathcal{H}$: (i) invariantly realizable setting, where there exists a $(\mathcal{G}, \mathcal{D}_{\mathcal{X}})$-invariant hypothesis $h^* \in \mathcal{H}$ with $\mathrm{err}_{\mathcal{D}}(h^*) = 0$; (ii) relaxed realizable setting, where there exists a (not necessarily $(\mathcal{G}, \mathcal{D}_{\mathcal{X}})$-invariant) hypothesis $h^* \in \mathcal{H}$ with $\mathrm{err}_{\mathcal{D}}(h^*) = 0$; and (iii) agnostic setting, where there might not exist a hypothesis in $\mathcal{H}$ with zero error. To understand the difference among the three settings, here is an example.

**Example 1.** *Consider $\mathcal{X} = \{\pm 1, \pm 2\}$, $\mathcal{G} = \{e, -e\}$ being the group generated by flipping the sign (i.e., $\mathcal{G}x = \{x, -x\}$), and the data distribution $\mathcal{D}$ being the uniform distribution over $\{(1, 0), (2, 0)\}$. If $\mathcal{H} = \{h(\cdot) = 0\}$ contains only the all-zero function, then it is in setting (i) as $h(\cdot) = 0$ is $(\mathcal{G}, \mathcal{D}_{\mathcal{X}})$-invariant and $\mathrm{err}_{\mathcal{D}}(h) = 0$; If $\mathcal{H} = \{\mathbb{1}[x < 0]\}$ contains only the hypothesis predicting $\{-1, -2\}$ as 1 and $\{1, 2\}$ as 0, then it is in setting (ii) as $\mathbb{1}[x < 0]$ is not $(\mathcal{G}, \mathcal{D}_{\mathcal{X}})$-invariant but $\mathrm{err}_{\mathcal{D}}(\mathbb{1}[x < 0]) = 0$; If $\mathcal{H} = \{\mathbb{1}[x > 0]\}$, it is in setting (iii) as no hypothesis in $\mathcal{H}$ has zero error.*

The following definitions formalize the notion of PAC learning in the three settings.

**Definition 1** (Invariantly realizable PAC learnability). *For any $\epsilon, \delta \in (0,1)$, the sample complexity of invariantly realizable $(\epsilon, \delta)$-PAC learning of $\mathcal{H}$ with respect to transformation group $\mathcal{G}$, denoted $\mathcal{M}_{\mathrm{INV}}(\epsilon, \delta; \mathcal{H}, \mathcal{G})$, is defined as the smallest $m \in \mathbb{N}$ for which there exists a learning rule $\mathcal{A}$ such that, for every $\mathcal{G}$-invariant data distribution $\mathcal{D}$ over $\mathcal{X} \times \mathcal{Y}$ where there exists a $\underline{(\mathcal{G}, \mathcal{D}_{\mathcal{X}})\text{-invariant}}$ predictor $h^* \in \mathcal{H}$ with zero error, $\mathrm{err}_{\mathcal{D}}(h^*) = 0$, with probability at least $1 - \delta$ over $S \sim \mathcal{D}^m$,*

$$\mathrm{err}_{\mathcal{D}}(\mathcal{A}(S)) \le \epsilon\,.$$

*If no such $m$ exists, define $\mathcal{M}_{\mathrm{INV}}(\epsilon, \delta; \mathcal{H}, \mathcal{G}) = \infty$. We say that $\mathcal{H}$ is PAC learnable in the invariantly realizable setting with respect to transformation group $\mathcal{G}$ if $\forall \epsilon, \delta \in (0,1)$, $\mathcal{M}_{\mathrm{INV}}(\epsilon, \delta; \mathcal{H}, \mathcal{G})$ is finite. For any algorithm $\mathcal{A}$, denote by $\mathcal{M}_{\mathrm{INV}}(\epsilon, \delta; \mathcal{H}, \mathcal{G}, \mathcal{A})$ the sample complexity of $\mathcal{A}$.*

**Definition 2** (Relaxed realizable PAC learnability). *For any $\epsilon, \delta \in (0,1)$, the sample complexity of relaxed realizable $(\epsilon, \delta)$-PAC learning of $\mathcal{H}$ with respect to transformation group $\mathcal{G}$, denoted $\mathcal{M}_{\mathrm{RE}}(\epsilon, \delta; \mathcal{H}, \mathcal{G})$, is defined as the smallest $m \in \mathbb{N}$ for which there exists a learning rule $\mathcal{A}$ such that, for every $\mathcal{G}$-invariant data distribution $\mathcal{D}$ over $\mathcal{X} \times \mathcal{Y}$ where there exists a predictor $h^* \in \mathcal{H}$ with zero error, $\mathrm{err}_{\mathcal{D}}(h^*) = 0$, with probability at least $1 - \delta$ over $S \sim \mathcal{D}^m$,*

$$\mathrm{err}_{\mathcal{D}}(\mathcal{A}(S)) \le \epsilon\,.$$

*If no such $m$ exists, define $\mathcal{M}_{\mathrm{RE}}(\epsilon, \delta; \mathcal{H}, \mathcal{G}) = \infty$. We say that $\mathcal{H}$ is PAC learnable in the relaxed realizable setting with respect to transformation group $\mathcal{G}$ if $\forall \epsilon, \delta \in (0,1)$, $\mathcal{M}_{\mathrm{RE}}(\epsilon, \delta; \mathcal{H}, \mathcal{G})$ is finite. For any algorithm $\mathcal{A}$, denote by $\mathcal{M}_{\mathrm{RE}}(\epsilon, \delta; \mathcal{H}, \mathcal{G}, \mathcal{A})$ the sample complexity of $\mathcal{A}$.*

**Definition 3** (Agnostic PAC learnability). *For any $\epsilon, \delta \in (0,1)$, the sample complexity of agnostic $(\epsilon, \delta)$-PAC learning of $\mathcal{H}$ with respect to transformation group $\mathcal{G}$, denoted $\mathcal{M}_{\mathrm{AG}}(\epsilon, \delta; \mathcal{H}, \mathcal{G})$, is defined as the smallest $m \in \mathbb{N}$ for which there exists a learning rule $\mathcal{A}$ such that, for every $\mathcal{G}$-invariant data distribution $\mathcal{D}$ over $\mathcal{X} \times \mathcal{Y}$, with probability at least $1 - \delta$ over $S \sim \mathcal{D}^m$,*

$$\mathrm{err}_{\mathcal{D}}(\mathcal{A}(S)) \le \inf_{h \in \mathcal{H}} \mathrm{err}_{\mathcal{D}}(h) + \epsilon\,.$$

*If no such $m$ exists, define $\mathcal{M}_{\mathrm{AG}}(\epsilon, \delta; \mathcal{H}, \mathcal{G}) = \infty$. We say that $\mathcal{H}$ is PAC learnable in the agnostic setting with respect to transformation group $\mathcal{G}$ if $\forall \epsilon, \delta \in (0,1)$, $\mathcal{M}_{\mathrm{AG}}(\epsilon, \delta; \mathcal{H}, \mathcal{G})$ is finite.*

**Data augmentation** One main goal of this work is to analyze the sample complexity of data augmentation. When we talk of data augmentation (DA) as an algorithm, it actually means ERM over the augmented data. Specifically, given a fixed loss function $\mathcal{L}$ mapping a data set and a hypothesis to $[0,1]$, and a training set $S_{\mathrm{trn}}$, DA outputs an $h \in \mathcal{H}$ such that $h(x) = y$ for all $(x, y) \in \mathcal{G}S_{\mathrm{trn}}$ if there exists one; outputs a hypothesis $h \in \mathcal{H}$ with the minimal loss $\mathcal{L}(\mathcal{G}S_{\mathrm{trn}}, h)$ otherwise. When we say DA without specifying the loss function, it means DA w.r.t. an arbitrary loss function, which can be defined based on any probability measure on the transformation group.

To characterize sample complexities, we define two measures as follows.

**Definition 4** (VC dimension of orbits). *The VC dimension of orbits, denoted $\mathrm{VC}_{\mathrm{o}}(\mathcal{H}, \mathcal{G})$, is defined as the largest integer $k$ for which there exists a set $X = \{x_1, \ldots, x_k\} \subset \mathcal{X}$ such that their orbits are pairwise disjoint, i.e., $\mathcal{G}x_i \cap \mathcal{G}x_j = \emptyset, \forall i, j \in [k]$ and every labeling of $X$ is realized by a $\underline{(\mathcal{G}, X)\text{-invariant}}$ hypothesis in $\mathcal{H}$, i.e., $\forall y \in \{0,1\}^k$, there exists a $(\mathcal{G}, X)$-invariant hypothesis $h \in \mathcal{H}$ s.t. $h(x_i) = y_i, \forall i \in [k]$.*

**Definition 5** (VC dimension across orbits). *The VC dimension across orbits, denoted $\mathrm{VC}_{\mathrm{ao}}(\mathcal{H}, \mathcal{G})$, is defined as the largest integer $k$ for which there exists a set $X = \{x_1, \ldots, x_k\} \subset \mathcal{X}$ such that their orbits are pairwise disjoint, i.e., $\mathcal{G}x_i \cap \mathcal{G}x_j = \emptyset, \forall i, j \in [k]$ and every labeling of $X$ is realized by a hypothesis in $\mathcal{H}$, i.e., $\forall y \in \{0,1\}^k$, there exists a hypothesis $h \in \mathcal{H}$ s.t. $h(x_i) = y_i, \forall i \in [k]$.*

Let $\mathrm{VCdim}(\mathcal{H})$ denote the VC dimension of $\mathcal{H}$. By definition, it is direct to check that $\mathrm{VC}_{\mathrm{o}}(\mathcal{H}, \mathcal{G}) \le \mathrm{VC}_{\mathrm{ao}}(\mathcal{H}, \mathcal{G}) \le \mathrm{VCdim}(\mathcal{H})$. For any $\mathcal{H}, \mathcal{G}$ with $\mathrm{VC}_{\mathrm{o}}(\mathcal{H}, \mathcal{G}) = d$, we can supplement $\mathcal{H}$ to a new hypothesis class $\mathcal{H}'$ such that $\mathrm{VC}_{\mathrm{o}}(\mathcal{H}', \mathcal{G})$ is still $d$ while $\mathrm{VC}_{\mathrm{ao}}(\mathcal{H}', \mathcal{G})$ is as large as the total number of orbits with at least two instances, i.e., $\mathrm{VC}_{\mathrm{ao}}(\mathcal{H}', \mathcal{G}) = |\{\mathcal{G}x \,|\, |\mathcal{G}x| \ge 2, x \in \mathcal{X}\}|$. This can be done by supplementing $\mathcal{H}$ with all hypotheses predicting $\mathcal{G}x$ with two different labels for all $x$ with $|\mathcal{G}x| \ge 2$. Besides, for any $\mathcal{H}$ with $\mathrm{VCdim}(\mathcal{H}) = d$, we can construct a transformation group $\mathcal{G}$ to make all instances lie in one single orbit, which makes $\mathrm{VC}_{\mathrm{ao}}(\mathcal{H}, \mathcal{G}) \le 1$. Hence the gap among the three measures can be arbitrarily large. Here are a few examples for better understanding of the gaps.

**Example 2.** *Consider $\mathcal{X} = \{\pm 1, \pm 2, \ldots, \pm 2d\}$ for some $d > 0$, $\mathcal{H} = \{\mathbb{1}[x \in A] | A \subset [2d] \text{ and } |A| = d\}$ being the set of all hypotheses labeling exact $d$ elements from $[2d]$ as $1$ and $\mathcal{G} = \{e, -e\}$ being the group generated by flipping the sign. Then we have $\mathrm{VC_o}(\mathcal{H}, \mathcal{G}) = 0$ since for any $i \in [2d]$, there is no $(\mathcal{G}, \{i\})$-invariant hypothesis that can label $i$ as $1$ (which is due to the fact that $-i$ is labeled as $0$ by any hypothesis in $\mathcal{H}$). It is direct to check that $\mathrm{VC_{ao}}(\mathcal{H}, \mathcal{G}) = \mathrm{VCdim}(\mathcal{H}) = d$.*

**Example 3.** *Consider $\mathcal{X} = \{x \in \mathbb{R}^2 | \|x\|_2 = 1\}$ being a circle, the hypothesis class $\mathcal{H} = \{0, 1\}^{\mathcal{X}}$ being all labeling functions and $\mathcal{G}$ being all rotations (thus $\forall x, \mathcal{G}x = \mathcal{X}$). Then we have $\mathrm{VC_o}(\mathcal{H}, \mathcal{G}) = \mathrm{VC_{ao}}(\mathcal{H}, \mathcal{G}) = 1$ as there is only one orbit and $\mathrm{VCdim}(\mathcal{H}) = \infty$.*

**Example 4.** *Consider the natural data being $k$ upright images and the transformation set $\mathcal{G}$ is rotation by $0, 360/n, 2 \cdot 360/n, \ldots, (n-1) \cdot 360/n$ degrees for some integer $n$. For an expressive hypothesis class $\mathcal{H}$ (e.g., neural networks) that can shatter all rotated versions of these images, we have $\mathrm{VCdim}(\mathcal{H}) = nk$ and $\mathrm{VC_o}(\mathcal{H}, \mathcal{G}) = \mathrm{VC_{ao}}(\mathcal{H}, \mathcal{G}) = k$. For a hypothesis class $\mathcal{H}'$ composed of all hypotheses labeling all upright images and their upside-down variations differently, we have $\mathrm{VCdim}(\mathcal{H}') = (n-1)k$, $\mathrm{VC_{ao}}(\mathcal{H}', \mathcal{G}) = k$ and $\mathrm{VC_o}(\mathcal{H}', \mathcal{G}) = 0$.*

## 3    Main results

We next present and discuss our main results.

- Invariantly realizable setting (Definition 1)
  - **DA "helps" but is not optimal.  The sample complexity of DA is characterized by $\mathrm{VC_{ao}}(\mathcal{H}, \mathcal{G})$.** For any $\mathcal{H}, \mathcal{G}$, DA can learn $\mathcal{H}$ with sample complexity $\widetilde{O}(\frac{\mathrm{VC_{ao}}(\mathcal{H}, \mathcal{G})}{\epsilon} + \frac{1}{\epsilon}\log\frac{1}{\delta})$, where $\widetilde{O}$ ignores log-factors of $\frac{\mathrm{VC_{ao}}(\mathcal{H}, \mathcal{G})}{\epsilon}$ (Theorem 1). For all $d > 0$, there exists $\mathcal{H}, \mathcal{G}$ with $\mathrm{VC_{ao}}(\mathcal{H}, \mathcal{G}) = d$ and $\mathrm{VC_o}(\mathcal{H}, \mathcal{G}) = 0$ such that DA needs $\Omega(\frac{d}{\epsilon})$ samples (Theorem 2).
  - **The optimal sample complexity is characterized by $\mathrm{VC_o}(\mathcal{H}, \mathcal{G})$.** For any $\mathcal{H}, \mathcal{G}$, we have $\Omega(\frac{\mathrm{VC_o}(\mathcal{H}, \mathcal{G})}{\epsilon} + \frac{1}{\epsilon}\log\frac{1}{\delta}) \leq \mathcal{M}_{\mathrm{INV}}(\epsilon, \delta; \mathcal{H}, \mathcal{G}) \leq O(\frac{\mathrm{VC_o}(\mathcal{H}, \mathcal{G})}{\epsilon}\log\frac{1}{\delta})$ (Theorem 4). We propose an algorithm achieving this upper bound based on 1-inclusion graphs, which does not distinguish between the original and transformed data. It is worth noting that the algorithm takes the invariance over the test point into account, which provides some theoretical justification for test-time adaptation such as Wang et al. (2021).
- Relaxed realizable setting (Definition 2)
  - **DA can "hurt".** DA belongs to the family of algorithms not distinguishing the original data from the transformed data. We show that the optimal sample complexity of this family is characterized by $\mu(\mathcal{H}, \mathcal{G})$ (see Definition 6) (Theorem 5), which can be arbitrarily larger than $\mathrm{VCdim}(\mathcal{H})$. This implies that for any $\mathcal{H}, \mathcal{G}$ with $\mu(\mathcal{H}, \mathcal{G}) > \mathrm{VCdim}(\mathcal{H})$, the sample complexity of DA is higher than that of ERM.
  - **The optimal sample complexity is characterized by $\mathrm{VC_{ao}}(\mathcal{H}, \mathcal{G})$.** For any $\mathcal{H}, \mathcal{G}$, we have $\Omega(\frac{\mathrm{VC_{ao}}(\mathcal{H}, \mathcal{G})}{\epsilon} + \frac{\log(1/\delta)}{\epsilon}) \leq \mathcal{M}_{\mathrm{RE}}(\epsilon, \delta; \mathcal{H}, \mathcal{G}) \leq \widetilde{O}(\frac{\mathrm{VC_{ao}}(\mathcal{H}, \mathcal{G})}{\epsilon} + \frac{1}{\epsilon}\log\frac{1}{\delta})$ (Theorem 7). We propose two algorithms achieving similar upper bounds, with one based on ERM and one based on 1-inclusion graphs. Both algorithms have to distinguish between the original and the transformed data.
  - **An adaptive algorithm interpolates between two settings.** We present an algorithm that adapts to different levels of invariance of the target function $h^*$, which achieves $\widetilde{O}(\frac{\mathrm{VC_{ao}}(\mathcal{H}, \mathcal{G})}{\epsilon})$ sample complexity in the relaxed realizable setting and $\widetilde{O}(\frac{\mathrm{VC_o}(\mathcal{H}, \mathcal{G})}{\epsilon})$ sample complexity in the invariantly realizable setting without knowing it (Theorem 9 in Appendix).
- Agnostic setting (Definition 3)
  - **The optimal sample complexity is characterized by $\mathrm{VC_{ao}}(\mathcal{H}, \mathcal{G})$.** For any $\mathcal{H}, \mathcal{G}$, $\mathcal{M}_{\mathrm{AG}}(\epsilon, \delta; \mathcal{H}, \mathcal{G}) = O\left(\frac{\mathrm{VC_{ao}}(\mathcal{H}, \mathcal{G})}{\epsilon^2}\log^2\left(\frac{\mathrm{VC_{ao}}(\mathcal{H}, \mathcal{G})}{\epsilon}\right) + \frac{1}{\epsilon^2}\log(\frac{1}{\delta})\right)$ (Theorem 8). Since $\mathcal{M}_{\mathrm{AG}}(\epsilon, \delta; \mathcal{H}, \mathcal{G}) \geq \mathcal{M}_{\mathrm{RE}}(\epsilon, \delta; \mathcal{H}, \mathcal{G})$, $\mathrm{VC_{ao}}(\mathcal{H}, \mathcal{G})$ characterizes the optimal sample complexity.

## 4    Invariantly Realizable setting

In this section, we discuss the results in the invariantly realizable setting (see Definition 1).

## 4.1 DA "helps" but is not optimal

We show that in the invariantly realizable setting, DA indeed "helps" to improve the sample complexity from $\widetilde{O}(\frac{\mathrm{VCdim}(\mathcal{H})}{\epsilon})$ (the sample complexity of ERM in standard PAC learning) to $\widetilde{O}(\frac{\mathrm{VC}_{\mathrm{ao}}(\mathcal{H},\mathcal{G})}{\epsilon})$. First, we have the following upper bound on the sample complexity of DA.

**Theorem 1.** *For any $\mathcal{H},\mathcal{G}$ with $\mathrm{VC}_{\mathrm{ao}}(\mathcal{H},\mathcal{G}) < \infty$, DA satisfies that $\mathcal{M}_{\mathrm{INV}}(\epsilon,\delta;\mathcal{H},\mathcal{G},\mathrm{DA}) = O(\frac{\mathrm{VC}_{\mathrm{ao}}(\mathcal{H},\mathcal{G})}{\epsilon} \log^3 \frac{\mathrm{VC}_{\mathrm{ao}}(\mathcal{H},\mathcal{G})}{\epsilon} + \frac{1}{\epsilon} \log \frac{1}{\delta})$.*

Intuitively, for a set of instances in one orbit that can be labeled by $\mathcal{H}$ in multiple ways, we only need to observe one instance from this orbit to learn the labels of all the instances by applying DA. Thus, DA helps to improve the accuracy. The detailed proof is deferred to Appendix A. However, DA does not fully exploit the transformation invariances as it only utilizes the invariances of the training set. Hence, DA does not perform optimally in presence of the transformation invariances. In fact, besides DA, all proper learners (i.e., learners outputting a hypothesis in $\mathcal{H}$) have the same problem.

**Theorem 2.** *For any $d > 0$, there exists a hypothesis class $\mathcal{H}_d$ and a group $\mathcal{G}_d$ with $\mathrm{VC}_{\mathrm{ao}}(\mathcal{H}_d,\mathcal{G}_d) = d$ and $\mathrm{VC}_{\mathrm{o}}(\mathcal{H}_d,\mathcal{G}_d) = 0$ such that $\mathcal{M}_{\mathrm{INV}}(\epsilon,\frac{1}{9};\mathcal{H}_d,\mathcal{G}_d,\mathcal{A}) = \Omega(\frac{d}{\epsilon})$ for any proper learner $\mathcal{A}$, including DA and standard ERM.*

The theorem shows that DA is sub-optimal as we will show that the optimal sample complexity is characterized by $\mathrm{VC}_{\mathrm{o}}(\mathcal{H},\mathcal{G})$ in Theorem 4. We provide an idea of the construction here and defer the detailed proof to Appendix B. Consider the $\mathcal{X}, \mathcal{H}$ and $\mathcal{G}$ in Example 2. Pick the target function $\mathbb{1}[x \in A]$ uniformly at random from $\mathcal{H}$ and let the data distribution only put probability mass on points in $[2d] \setminus A$, the orbits of which are labeled as $0$ by the target function. Then any proper learner must predict $d$ unobserved examples of $[2d]$ as $1$, which leads to high error if the learner observes fewer than $d/2$ examples. Theorem 2 also implies that for any hypothesis class including $\mathcal{H}$ as a subset, there exists a DA learner (i.e., a proper learner fitting the augmented data) whose sample complexity is $\Omega(\frac{d}{\epsilon})$.

Theorem 1 shows that the sample complexity of DA is $\widetilde{O}(\frac{\mathrm{VC}_{\mathrm{ao}}(\mathcal{H},\mathcal{G})}{\epsilon})$, better than that of ERM in standard PAC learning, $\widetilde{O}(\frac{\mathrm{VCdim}(\mathcal{H})}{\epsilon})$. This is insufficient to show that DA outperforms ERM as it might be possible that ERM can also achieve better sample complexity in presence of transformation invariances. To illustrate that DA indeed "helps", we show that any algorithm without exploiting the transformation invariances still requires sample complexity of $\Omega(\frac{\mathrm{VCdim}(\mathcal{H})}{\epsilon})$.

**Theorem 3.** *For any $\mathcal{H}$, there exists a group $\mathcal{G}$ with $\mathrm{VC}_{\mathrm{ao}}(\mathcal{H},\mathcal{G}) \leq 5$ s.t. $\mathcal{M}_{\mathrm{INV}}(\epsilon,\delta;\mathcal{H},\mathcal{G},\mathcal{A}) = \Omega(\frac{\mathrm{VCdim}(\mathcal{H})}{\epsilon} + \frac{1}{\epsilon} \log \frac{1}{\delta})$ for any algorithm $\mathcal{A}$ not given any information about $\mathcal{G}$ (e.g., ERM).*

The basic idea is that, given a set of $k$ instances that can be shattered by $\mathcal{H}$ for some $k > 0$, $\mathcal{G}$ is uniformly at random picked from a set of $2^k$ groups, each of which partitions the set into two orbits in a different way. If given $\mathcal{G}$, the algorithm only need to observe one instance in each orbit to learn the labels of all $k$ instances. If not, the algorithm can only randomly guess the label of every unobserved instance. The detailed construction is included in Appendix C.

## 4.2 The optimal algorithm

We show that the optimal sample complexity is characterized by $\mathrm{VC}_{\mathrm{o}}(\mathcal{H},\mathcal{G})$.

**Theorem 4.** *For any $\mathcal{H},\mathcal{G}$ with $\mathrm{VC}_{\mathrm{o}}(\mathcal{H},\mathcal{G}) < \infty$, we have $\Omega(\frac{\mathrm{VC}_{\mathrm{o}}(\mathcal{H},\mathcal{G})}{\epsilon} + \frac{1}{\epsilon} \log \frac{1}{\delta}) \leq \mathcal{M}_{\mathrm{INV}}(\epsilon,\delta;\mathcal{H},\mathcal{G}) \leq O(\frac{\mathrm{VC}_{\mathrm{o}}(\mathcal{H},\mathcal{G})}{\epsilon} \log \frac{1}{\delta})$.*

Our algorithm is based on the 1-inclusion-graph predictor by Haussler et al. (1994). Given hypothesis class $H$ and instance space $X = \{x_1, \ldots, x_t\}$, the classical 1-inclusion-graph consists of vertices $\{h_{|X} | h \in H\}$, which are labelings of $X$ realized by $H$, and two vertices are connected by an edge if and only if they differ at the labeling of exactly a single $x_i \in X$. Haussler et al. (1994) shows that the edges can be oriented such that each vertex has in-degree at most $\mathrm{VCdim}(H)$. This orientation can be translated to a prediction rule. Specifically, for any $i \in [t]$, given the labels of all instances in $X$ except $x_i$, if there are two hypotheses $h, h' \in H$ such that their labelings are consistent with the labels of $X \setminus \{x_i\}$ and different at $x_i$, then $h_{|X}, h'_{|X}$ are two vertices in the graph and we predict

the label of $x_i$ as the edge between $h_{|X}, h'_{|X}$ is oriented against. The average leave-one-out-error is upper bounded by $\frac{\text{VCdim}(H)}{t}$.

**Lemma 1** (Theorem 2.3 of Haussler et al. (1994)). *For any hypothesis class $H$ and instance space $X$ with $\text{VCdim}(H) < \infty$, there is a function $Q : (X \times \mathcal{Y})^* \times X \mapsto \mathcal{Y}$ such that, for any $t \in \mathbb{N}$ and sample $\{(x_1, y_1), \ldots, (x_t, y_t)\}$ that is realizable w.r.t. $H$,*

$$\frac{1}{t!} \sum_{\sigma \in \text{Sym}(t)} \mathbb{1}[Q(\{x_{\sigma(i)}, y_{\sigma(i)}\}_{i \in [t-1]}, x_{\sigma(t)}) \neq y_{\sigma(t)}] \leq \frac{\text{VCdim}(H)}{t}, \tag{1}$$

*where $\text{Sym}(t)$ denotes the symmetric group on $[t]$. The function $Q$ can be constructed by a 1-inclusion-graph predictor.*

Denote by $Q_{H,X}$ the function guaranteed by Eq (1) for hypothesis class $H$ and instance space $X$. For any $t \in \mathbb{N}$ and $S = \{(x_1, y_1), \ldots, (x_t, y_t)\}$, let $X_S$ denote the set of different elements in $S_{\mathcal{X}}$. Define $\mathcal{H}(X_S) := \{h_{|X_S}|h \in \mathcal{H} \text{ is } (\mathcal{G}, X_S)\text{-invariant}\}$ being the set of all possible $(\mathcal{G}, X_S)$-invariant labelings of $X_S$. We then define our algorithm $\mathcal{A}(S)$ by letting $\mathcal{A}(S)(x) = Q_{\mathcal{H}(X_S \cup \{x\}), X_S \cup \{x\}}(S, x)$ if $\mathcal{H}(X_S \cup \{x\}) \neq \emptyset$ and predicting arbitrarily if $\mathcal{H}(X_S \cup \{x\}) = \emptyset$. That is to say, $\mathcal{A}(S)$ needs to construct a function $Q$ for every test example. Given any test example, this 1-inclusion-graph-based algorithm takes into account whether the prediction can be invariant over the whole orbit of the test example and thus benefits from the invariance of test examples. This can provide some theoretical justification for test-time adaptation such as Wang et al. (2021). By definition, we have $\text{VCdim}(\mathcal{H}(X_S \cup \{x\})) \leq \text{VC}_o(\mathcal{H}, \mathcal{G})$ for all $S$ and $x$. Then the expected error of $\mathcal{A}$ can be bounded by $\text{VC}_o(\mathcal{H}, \mathcal{G})$ through Lemma 1. We defer the details and the proof of Theorem 4 to Appendix D. Note that the results of Theorem 4 also apply to non-group transformations[2].

## 5 Relaxed realizable setting

In this section, we discuss the results in the relaxed realizable setting (see Definition 2). As we can see, DA belongs to the family of algorithms not distinguishing between the original and transformed data. In Section 5.1, we provide a tight characterization $\mu(\mathcal{H}, \mathcal{G})$ (Definition 6) on the sample complexity of this family algorithms. This implies that when $\mu(\mathcal{H}, \mathcal{G}) > \text{VCdim}(\mathcal{H})$, there exists a distribution s.t. DA performs worse than ERM. We then show that there exists $\mathcal{H}, \mathcal{G}$ such that $\mu(\mathcal{H}, \mathcal{G}) > \text{VCdim}(\mathcal{H})$ and the gap can be arbitrarily large. In Section 5.2, we provide two optimal algorithms, both of which have to distinguish between the original and transformed data.

### 5.1 DA can even "hurt"

In the invariantly realizable setting, the optimal algorithm based on 1-inclusion graphs does not need to distinguish between the original and transformed data since $\mathcal{H}(X_S \cup \{x\})$ in the algorithm is fully determined by the augmented data. However, in the relaxed realizable setting, distinguishing between the original and transformed data is crucial. In the following, we will provide a characterization of the sample complexity of algorithms not distinguishing between the original and transformed data, including DA. Such a characterization induces a sufficient condition when DA "hurts".

Let $\mathcal{M}_{\text{DA}}(\epsilon, \delta; \mathcal{H}, \mathcal{G})$ be the smallest integer $m$ for which there exists a learning rule $\mathcal{A}$ such that for every $\mathcal{G}$-invariant data distribution $\mathcal{D}$, with probability at least $1 - \delta$ over $S_{\text{trn}} \sim \mathcal{D}^m$, $\text{err}_{\mathcal{D}}(\mathcal{A}(\mathcal{G}S_{\text{trn}})) \leq \epsilon$. The quantity $\mathcal{M}_{\text{DA}}(\epsilon, \delta; \mathcal{H}, \mathcal{G})$ is the optimal sample complexity achievable if algorithms can only access the augmented data without knowing the original training set. In standard PAC learning, the optimal sample complexity can be characterized by the maximum density of any subgraph of the 1-inclusion graphs, which is actually equal to the VC dimension (Haussler et al., 1994; Daniely and Shalev-Shwartz, 2014). Analogously, we characterize $\mathcal{M}_{\text{DA}}(\epsilon, \delta; \mathcal{H}, \mathcal{G})$ based on a variant of 1-inclusion graphs, which is constructed as follows.

In the 1-inclusion graph for standard PAC learning (Haussler et al., 1994), given any sequence of instances $\mathbf{x} = (x_1, \ldots, x_t)$, the vertices are labelings of $\mathbf{x}$ and two vertices are connected iff. they are different at only one instance in $\mathbf{x}$ and this instance appears once. In our setting, the input is a multiset of labeled orbits and an unlabeled test instance, hence the vertices are pairs of labelings of orbits

---

[2]In this case, we only assume that $\mathcal{G}$ contains the identity element.

and unlabeled instances. Specifically, for any $t \in \mathbb{N}$, given a multi-set of orbits $\boldsymbol{\phi} = \{\phi_1, \ldots, \phi_t\}$ of some unknown original data, a labeling $\mathbf{f} \in \mathcal{Y}^t$ is possible iff. there exists a sequence of instances $\mathbf{x} = (x_1, \ldots, x_t) \in \prod_{i=1}^{t} \phi_i$ and a hypothesis $h \in \mathcal{H}$ such that $h_{|\mathbf{x}} = \mathbf{f}$ and that instances in the same orbit are labeled the same, i.e., $(\mathcal{G}x_i \times \{1 - f_i\}) \cap (\{(x_j, f_j)\}_{j \in [t]}) = \emptyset$ for all $i \in [t]$. We denote the set of all possible labelings of $\boldsymbol{\phi}$ by

$$\Pi_{\mathcal{H}}(\boldsymbol{\phi}) := \{\mathbf{f} \in \mathcal{Y}^t | \exists h \in \mathcal{H}, \exists \mathbf{x} \in \prod_{i=1}^{t} \phi_i, h_{|\mathbf{x}} = \mathbf{f} \text{ and } \cup_{i \in [t]} (\mathcal{G}x_i \times \{1 - f_i\}) \cap \{(x_j, f_j) | j \in [t]\} = \emptyset\}. \tag{2}$$

Denote the set of all such sequences $\mathbf{x} \in \prod_{i=1}^{t} \phi_i$ of instances that can be labeled as $\mathbf{f}$ by

$$\mathcal{U}_{\mathbf{f}}(\boldsymbol{\phi}) := \{\mathbf{x} \in \prod_{i=1}^{t} \phi_i | \exists h \in \mathcal{H}, h_{|\mathbf{x}} = \mathbf{f} \text{ and } \cup_{i \in [t]} (\mathcal{G}x_i \times \{1 - f_i\}) \cap \{(x_j, f_j) | j \in [t]\} = \emptyset\}.$$

Denote the set of all pairs of labeling and its corresponding instance sequence by

$$B(\mathcal{H}, \mathcal{G}, \boldsymbol{\phi}) := \cup_{\mathbf{f} \in \Pi_{\mathcal{H}}(\boldsymbol{\phi})} \{\mathbf{f}\} \times \mathcal{U}_{\mathbf{f}}(\boldsymbol{\phi}). \tag{3}$$

For any $(\mathbf{f}, \mathbf{x}) \in B(\mathcal{H}, \mathcal{G}, \boldsymbol{\phi})$, $\mathbf{x}$ is a candidate of original data and $\mathbf{f}$ is a candidate of labeling of $\mathbf{x}$. Now we define a graph $G_{\mathcal{H}, \mathcal{G}}(\boldsymbol{\phi})$, where the vertices are all pairs of labeling $\mathbf{f} \in \Pi_{\mathcal{H}}(\boldsymbol{\phi})$ and an element in a instance sequence corresponding to $\mathbf{f}$. Formally, the vertex set is

$$V = \{(\mathbf{f}, x_i) | (\mathbf{f}, \mathbf{x}) \in B(\mathcal{H}, \mathcal{G}, \boldsymbol{\phi}), i \in [t]\}.$$

For every two vertices $(\mathbf{f}, x)$ and $(\mathbf{g}, z)$, they are connected if and only if (i) $x = z$; (ii) there exists $j \in [t]$ such that $x \in \phi_j$, $f_i = g_i, \forall i \neq j$ and $f_j \neq g_j$; and (iii) $\phi_j$ only appear once in $\boldsymbol{\phi}$. Each edge can be represented by $e = \{\mathbf{f}, \mathbf{g}, x\}$ and we denote $E$ the edge set. If an edge $e = \{\mathbf{f}, \mathbf{g}, x\}$ exists, the edge could be recovered given only $\mathbf{f}, x$ or $\mathbf{g}, x$, and thus, we also denote by $e(\mathbf{f}, x) = e(\mathbf{g}, x) = \{\mathbf{f}, \mathbf{g}, x\}$. Any algorithm accessing only the augmented data corresponds to an orientation of edges in the graph we constructed, which leads to the following definition.

**Definition 6.** *Let $w : E \times \Pi_{\mathcal{H}}(\boldsymbol{\phi}) \mapsto [0, 1]$ be a mapping such that for every $e = \{\mathbf{f}, \mathbf{g}, x\}$, $w(e, \mathbf{f}) + w(e, \mathbf{g}) = 1$ and $w(e, \mathbf{h}) = 0$ if $\mathbf{h} \notin e$ and let $W$ be the set of all such mappings. Note that $w$ actually defines a randomized orientation of each edge in graph $G_{\mathcal{H}, \mathcal{G}}(\boldsymbol{\phi})$: the edge $e = \{\mathbf{f}, \mathbf{g}, x\}$ is oriented towards vertex $(\mathbf{f}, x)$ with probability $w(e, \mathbf{f})$. For any $(\mathbf{f}, \mathbf{x}) \in B(\mathcal{H}, \mathcal{G}, \boldsymbol{\phi})$, it corresponds to a cluster of vertices $\{(\mathbf{f}, x_i) | i \in [t]\}$ in $G_{\mathcal{H}, \mathcal{G}}(\boldsymbol{\phi})$ and $\sum_{i \in [t]: \exists e \in E, \{\mathbf{f}, x_i\} \subset e} w(e(\mathbf{f}, x_i), \mathbf{f})$ is the expected in-degree of the cluster. Let $\Delta(B(\mathcal{H}, \mathcal{G}, \boldsymbol{\phi}))$ denote the set of all distributions over $B(\mathcal{H}, \mathcal{G}, \boldsymbol{\phi})$. For any $P \in \Delta(B(\mathcal{H}, \mathcal{G}, \boldsymbol{\phi}))$, we define*

$$\mu(\mathcal{H}, \mathcal{G}, \boldsymbol{\phi}, P) := \min_{w \in W} \mathbb{E}_{(\mathbf{f}, \mathbf{x}) \sim P} \left[ \sum_{i \in [t]: \exists e \in E, \{\mathbf{f}, x_i\} \subset e} w(e(\mathbf{f}, x_i), \mathbf{f}) \right]. \tag{4}$$

*By taking the supremum over $P$, we define $\mu(\mathcal{H}, \mathcal{G}, \boldsymbol{\phi}) := \sup_{P \in \Delta(B(\mathcal{H}, \mathcal{G}, \boldsymbol{\phi}))} \mu(\mathcal{H}, \mathcal{G}, \boldsymbol{\phi}, P)$. By taking supremum over $\boldsymbol{\phi}$, we define $\mu(\mathcal{H}, \mathcal{G}, t) := \sup_{\boldsymbol{\phi}: |\boldsymbol{\phi}| = t} \mu(\mathcal{H}, \mathcal{G}, \boldsymbol{\phi})$ and*

$$\mu(\mathcal{H}, \mathcal{G}) := \sup_{t \in \mathbb{N}} \mu(\mathcal{H}, \mathcal{G}, t). \tag{5}$$

**Theorem 5.** *For any $\mathcal{H}, \mathcal{G}$, $\mathcal{M}_{\mathrm{DA}}(\epsilon, \delta; \mathcal{H}, \mathcal{G})$ satisfies the following bounds:*

- *For all $t \geq 2$ with $\mu(\mathcal{H}, \mathcal{G}, t) < \infty$, $\mathcal{M}_{\mathrm{DA}}(\epsilon, \frac{\mu(\mathcal{H}, \mathcal{G}, t)}{16(t-1)}; \mathcal{H}, \mathcal{G}) = \Omega(\frac{\mu(\mathcal{H}, \mathcal{G}, t)}{\epsilon})$. This implies that if $\mu(\mathcal{H}, \mathcal{G}) < \infty$, there exists a constant $c$ dependent on $\mathcal{H}, \mathcal{G}$ s.t. $\mathcal{M}_{\mathrm{DA}}(\epsilon, c; \mathcal{H}, \mathcal{G}) = \Omega(\frac{\mu(\mathcal{H}, \mathcal{G})}{\epsilon})$.*

- *For all $t$ with $\frac{1}{6} \leq \frac{\mu(\mathcal{H}, \mathcal{G}, t)}{t} \leq \frac{1}{3}$, $\mathcal{M}_{\mathrm{DA}}(\epsilon, \delta; \mathcal{H}, \mathcal{G}) = O(\frac{\mu(\mathcal{H}, \mathcal{G}, t)}{\epsilon} \log^2 \frac{\mu(\mathcal{H}, \mathcal{G}, t)}{\epsilon} + \frac{1}{\epsilon} \log \frac{1}{\delta})$.*

- *If $\mu(\mathcal{H}, \mathcal{G}) < \infty$, $\mathcal{M}_{\mathrm{DA}}(\epsilon, \delta; \mathcal{H}, \mathcal{G}) = O(\frac{\mu(\mathcal{H}, \mathcal{G}) \log(1/\delta)}{\epsilon})$.*

Theorem 5 implies that when $\mu(\mathcal{H}, \mathcal{G}) > \mathrm{VCdim}(\mathcal{H})$, there exists a distribution such that any algorithm not differentiating between the original and transformed data performs worse than simply applying ERM over the original data. We defer the proof of Theorem 5 to Appendix E. As we can see, the definition of $\mu(\mathcal{H}, \mathcal{G})$ is not intuitive and it might be difficult to calculate $\mu(\mathcal{H}, \mathcal{G})$ as well as to further determine when $\mu(\mathcal{H}, \mathcal{G}) > \mathrm{VCdim}(\mathcal{H})$. We introduce a new dimension as follows, which lower bounds $\mu(\mathcal{H}, \mathcal{G})$ and is easier to calculate.

**Definition 7** (VC Dimension of orbits generated by $\mathcal{H}$). *The VC dimension of orbits generated by $\mathcal{H}$, denoted $\dim(\mathcal{H}, \mathcal{G})$, is defined as the largest integer $k$ for which there exists a set $X = \{x_1, \ldots, x_k\} \subset \mathcal{X}$ such that (i) their orbits $\phi = \{\phi_1, \ldots, \phi_k\}$ are pairwise disjoint, (ii) $\Pi_{\mathcal{H}}(\phi) = \mathcal{Y}^k$ (defined in Eq (2)) and (iii) there exists a set $B = \{(\mathbf{f}, \mathbf{x_f})\}_{\mathbf{f} \in \mathcal{Y}^k} \subset B(\mathcal{H}, \mathcal{G}, \phi)$ (defined in Eq (3)) such that $\mathbf{f} \oplus \mathbf{g} = \mathbf{e}_i$ implies $x_{\mathbf{f},i} = x_{\mathbf{g},i}$ for all $i \in [k], \mathbf{f}, \mathbf{g} \in \mathcal{Y}^k$.*

**Theorem 6.** *For any $\mathcal{H}, \mathcal{G}$, $\mu(\mathcal{H}, \mathcal{G}) \geq \dim(\mathcal{H}, \mathcal{G})/2$.*

The proof is included in Appendix F. Through this dimension, we claim that the gap between $\mu(\mathcal{H}, \mathcal{G})$ and $\mathrm{VCdim}(\mathcal{H})$ can be arbitrarily large. In the following, we give an example of $\mathcal{H}, \mathcal{G}$ with $\dim(\mathcal{H}, \mathcal{G}) \gg \mathrm{VCdim}(\mathcal{H})$, which cannot be learned by DA but can be easily learned by ERM.

**Example 5.** *For any $d > 0$, let $\mathcal{X} = \{\pm 1\} \times \{\mathbf{e}_i | i \in [2d]\} \subset \mathbb{R}^{2d+1}$, $\mathcal{H} = \{x_1 > 0, x_1 \leq 0\}$ and $\mathcal{G} = \{I_{2d+1}, I_{2d+1} - 2\mathrm{diag}(\mathbf{e}_1)\}$ (i.e., the cyclic group generated by flipping the sign of $x_1$). It is easy to check that $\mathrm{VCdim}(\mathcal{H}) = 1$. Let $X = \{(1, \mathbf{e}_1), (1, \mathbf{e}_2), \ldots, (1, \mathbf{e}_{2d})\}$ and then the orbits generated from $X$ are $\phi = \{\{(-1, \mathbf{e}_i), (1, \mathbf{e}_i)\} | i \in [2d]\}\}$. For every labeling $\mathbf{f} \in \mathcal{Y}^{2d}$, if $\sum_{i \in [2d]} f_i$ is odd, let $\mathbf{x_f} = ((2f_i - 1, \mathbf{e}_i))_{i=1}^{2d}$; if $\sum_{i \in [2d]} f_i$ is even, let $\mathbf{x}_f = ((1 - 2f_i, \mathbf{e}_i))_{i=1}^{2d}$. It is direct to check that $(\mathbf{f}, \mathbf{x_f}) \in B(\mathcal{H}, \mathcal{G}, \phi)$ for all $\mathbf{f} \in \mathcal{Y}^{2d}$. Then for all $i \in [2d]$, $\mathbf{f} \oplus \mathbf{g} = \mathbf{e}_i$ implies $x_{\mathbf{f},i} = x_{\mathbf{g},i}$. Hence, $\dim(\mathcal{H}, \mathcal{G}) = 2d$, where $d$ can be an arbitrary positive integer. According to Theorem 6 we have $\mu(\mathcal{H}, \mathcal{G}) \geq d$.*

The above example can be interpreted in a vision scenario. Let's consider an example of classifying land birds versus water birds. The natural data is $2d$ images of land birds with land background and water birds with water background. The transformation set is composed of keeping the current background and changing the background from land (water) to water (land). Consider simple hypotheses depending on backgrounds only. Specifically, $\mathcal{H} = \{h_1, h_2\}$ with $h_1$ predicting all images with water background as water birds and $h_2$ predicting all images with water background as land birds. Let the data distribution be the uniform distribution over all the original images. Then given any training data, $h_1$ and $h_2$ have the same empirical loss on the augmented training data. Thus, for any unobserved image, DA will make a mistake with constant probability. Hence DA requires at least $\Omega(d)$ sample complexity. It is direct to check that standard ERM only needs one labeled instance to achieve zero error.

**Open question:** It is unclear whether $\mu(\mathcal{H}, \mathcal{G})$ is upper bounded by $\dim(\mathcal{H}, \mathcal{G})$. If true, then we can tightly characterize $\mathcal{M}_{\mathrm{DA}}(\epsilon, \delta; \mathcal{H}, \mathcal{G})$ by $\dim(\mathcal{H}, \mathcal{G})$.

### 5.2 The optimal algorithms

Different from the invariantly realizable setting, the optimal sample complexity in the relaxed realizable setting is characterized by $\mathrm{VC}_{\mathrm{ao}}(\mathcal{H}, \mathcal{G})$. The optimal (up to log-factors) sample complexity can be achieved by another variant of 1-inclusion-graph predictor. Besides, we propose an ERM-based algorithm, called ERM-INV (see Appendix G for details), achieving the similar guarantee.

**Theorem 7.** *For any $\mathcal{H}, \mathcal{G}$ with $\mathrm{VC}_{\mathrm{ao}}(\mathcal{H}, \mathcal{G}) < \infty$, we have $\Omega(\frac{\mathrm{VC}_{\mathrm{ao}}(\mathcal{H},\mathcal{G})}{\epsilon} + \frac{\log(1/\delta)}{\epsilon}) \leq \mathcal{M}_{\mathrm{RE}}(\epsilon, \delta; \mathcal{H}, \mathcal{G}) \leq O\left(\min\left(\frac{\mathrm{VC}_{\mathrm{ao}}(\mathcal{H},\mathcal{G})}{\epsilon} \log^3 \frac{\mathrm{VC}_{\mathrm{ao}}(\mathcal{H},\mathcal{G})}{\epsilon} + \frac{1}{\epsilon} \log \frac{1}{\delta}, \frac{\mathrm{VC}_{\mathrm{ao}}(\mathcal{H},\mathcal{G})}{\epsilon} \log \frac{1}{\delta}\right)\right)$.*

We defer the details of algorithms and the proof of Theorem 7 to Appendix G. Usually, ERM-INV is more efficient than the 1-inclusion-graph predictor. But the 1-inclusion-graph predictor as well as the lower bound can apply to non-group transformations. Another advantage of 1-inclusion-graph predictor is allowing us to design an adaptive framework which automatically adjusts to different levels of invariance of $h^*$. Specifically, for any hypothesis $h \in \mathcal{H}$, we say $h$ is $(1 - \eta)$-invariant over the distribution $\mathcal{D}_{\mathcal{X}}$ for some $\eta \in [0, 1]$ if $\mathbb{P}_{x \sim \mathcal{D}_{\mathcal{X}}}(\exists x' \in \mathcal{G}x, h(x') \neq h(x)) = \eta$. When $\eta(h^*) = 0$, it degenerates into the invariantly realizable setting, which implies that we can achieve better bounds when $\eta(h^*)$ is smaller. We propose an adaptive algorithm with sample complexity dependent on $\eta(h^*)$ and the details are included in Appendix I.1.

## 6 Agnostic setting

In the agnostic setting (Definition 3), $\inf_{h \in \mathcal{H}} \mathrm{err}(h)$ is possibly non-zero. Different from the agnostic setting in the standard PAC learning allowing probabilistic labels, our problem is limited to

deterministic labels because we assume that the data distribution is $\mathcal{G}$-invariant, i.e., there exists a $(\mathcal{G}, \mathcal{D}_{\mathcal{X}})$-invariant hypothesis $f^*$ (possibly not in $\mathcal{H}$) with $\mathrm{err}_{\mathcal{D}}(f^*) = 0$.

**Theorem 8.** *The sample complexity in the agnostic setting satisfies:*

- *For all $d > 0$, there exists $\mathcal{H}, \mathcal{G}$ with $\mathrm{VC}_{\mathrm{ao}}(\mathcal{H}, \mathcal{G}) = d$, $\mathcal{M}_{\mathrm{AG}}(\epsilon, 1/64; \mathcal{H}, \mathcal{G}) = \Omega(\frac{d}{\epsilon^2})$.*

- *For any $\mathcal{H}, \mathcal{G}$ with $\mathrm{VC}_{\mathrm{ao}}(\mathcal{H}, \mathcal{G}) < \infty$, $\mathcal{M}_{\mathrm{AG}}(\epsilon, \delta; \mathcal{H}, \mathcal{G}) = \widetilde{O}\left( \frac{\mathrm{VC}_{\mathrm{ao}}(\mathcal{H}, \mathcal{G})}{\epsilon^2} + \frac{1}{\epsilon^2} \log(\frac{1}{\delta}) \right)$.*

For upper bound, we show that ERM-INV achieves sample complexity $\widetilde{O}(\frac{\mathrm{VC}_{\mathrm{ao}}(\mathcal{H}, \mathcal{G})}{\epsilon^2})$. There is another way of achieving similar upper bound based on applying the reduction-to-realizable technique of David et al. (2016). Note that a direct combination of any reduction-to-realizable technique and any optimal algorithm in relaxed realizable setting does not work in our agnostic setting. This is because the relaxed realizable setting requires not only realizability, but also invariance in the support of the data distribution. For example, the reduction method of Hopkins et al. (2021) needs to run a realizable algorithm over a set labeled by each $h \in \mathcal{H}$, which might label two instances in the same orbit differently and make the realizable algorithm not well-defined. When combining the reduction method of David et al. (2016) and the 1-inclusion-graph-type algorithm, the similar problem also exists but can be fixed by predicting arbitrarily when the invariance property is not satisfied. For lower bound, According to Ben-David and Urner (2014), the sample complexity of agnostic PAC learning under deterministic labels is not fully determined by the VC dimension. Following the construction by Ben-David and Urner (2014), we provide an analogous lower bound in our setting. The algorithm details and the proofs are deferred to Appendix H. Analogous to the realizable setting, we provide one algorithm adapting to different levels of invariance of the optimal hypothesis in $\mathcal{H}$ in Appendix I.2. Similar to the results in the realizable settings, the lower bound and the 1-inclusion-graph predictor in the agnostic setting also apply to non-group transformations.

## 7   Discussion

**Definition of invariance under probabilistic labels** In this work, we model invariance by assuming that the data distribution is $\mathcal{G}$-invariant, which restricts the labels to be deterministic. It is unclear what "invariance under probabilistic labels" means. One option is assuming that the distribution of the labels is invariant over the orbits, $\Pr(y|gx) = \Pr(y|x)$ for all $g \in \mathcal{G}, x \in \mathcal{X}$. However, such a condition may not characterize invariance in real-world scenarios due to classes having different underlying distributions. For example, given a fuzzy image with probability $0.5$ being a car and $0.5$ being a tree, it is uncertain if the chance of this image being a car is still $0.5$ after rotation.

**The performance of DA under non-group transformations** Most results of DA and ERM-type algorithms only hold when the transformation set is a group. If we regard adversarial training as a special type of data augmentation through a ball around the natural data, then the transformation set is not a group. The appropriate way to formulate theoretical guarantees for DA under arbitrary transformations is still an open question.

## Acknowledgements

This work was supported in part by the National Science Foundation under grant CCF-1815011 and by the Defense Advanced Research Projects Agency under cooperative agreement HR00112020003. The views expressed in this work do not necessarily reflect the position or the policy of the Government and no official endorsement should be inferred. Approved for public release; distribution is unlimited.

We thank anonymous reviewers for their valuable suggestions. HS thanks Freda Shi for discussion on the application of DA and suggestions from an applied viewpoint.

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
