$\text{VCdim}(\mathcal{H}) = nk$ and $\text{VC}_\text{o}(\mathcal{H}, \mathcal{G}) = \text{VC}_\text{ao}(\mathcal{H}, \mathcal{G}) = k$. For a hypothesis class $\mathcal{H}'$ composed of all hypotheses labeling all upright images and their upside-down variations differently, we have $\text{VCdim}(\mathcal{H}') = (n - 1)k$, $\text{VC}_\text{ao}(\mathcal{H}', \mathcal{G}) = k$ and $\text{VC}_\text{

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

# A Proof of Theorem 1

*Proof.* Let $d = \text{VC}_{\text{ao}}(\mathcal{H}, \mathcal{G})$. According to the definition of DA and the invariantly realizable setting, given the input $S \sim \mathcal{D}^m$, the output of DA $\widehat{h} \in \mathcal{H}$ satisfies $\text{err}_{\mathcal{G}S}(\widehat{h}) = 0$, i.e., $\widehat{h}(x) = y$ for all $(x, y) \in \mathcal{G}S$. Consider two sets $S$ and $S'$ of $m$ i.i.d. samples drawn from the data distribution $\mathcal{D}$ each. We denote $A_S$ the event of $\{\exists h \in \mathcal{H}, \text{err}_{\mathcal{D}}(h) \geq \epsilon, \text{err}_{\mathcal{G}S}(h) = 0\}$ and $B_{S,S'}$ the event of $\{\exists h \in \mathcal{H}, \text{err}_{S'}(h) \geq \frac{\epsilon}{2}, \text{err}_{\mathcal{G}S}(h) = 0\}$. By Chernoff bound, we have $\Pr(B_{S,S'}) \geq \Pr(A_S) \cdot \Pr(B_{S,S'}|A_S) \geq \frac{1}{2}\Pr(A_S)$ when $m \geq \frac{8}{\epsilon}$. The sampling process of $S$ and $S'$ is equivalent to drawing $2m$ i.i.d. samples and then randomly partitioning into $S$ and $S'$ of $m$ each. For any fixed $S''$, for any $h$ with $\text{err}_{S'}(h) \geq \frac{\epsilon}{2}$ and $\text{err}_{\mathcal{G}S}(h) = 0$, if $h$ misclassifies some $(x, y) \in S''$, then all examples in the orbit of $x$, i.e., $\mathcal{G}\{(x,y)\} \cap S''$, must go to $S'$. Now to prove the theorem, we divide $S''$ into two categories in terms of the number of examples in each orbit. Let $R_1 = \{x \mid |\mathcal{G}x \cap S''_{\mathcal{X}}| \geq \log^2 m, x \in S''_{\mathcal{X}}\}$ and $R_2 = S''_{\mathcal{X}} \setminus R_1$. For any $h$ making at least $\frac{\epsilon m}{2}$ mistakes in $S''_{\mathcal{X}}$, $h$ either makes at least $\frac{\epsilon m}{4}$ mistakes in $R_1$ or makes at least $\frac{\epsilon m}{4}$ mistakes in $R_2$. Then let $\mathcal{H}_0 \subset \mathcal{H}$ denote the set of hypotheses making at least $\frac{\epsilon m}{2}$ mistakes in $S''_{\mathcal{X}}$ and divide $\mathcal{H}_0$ into two sub-classes as follows.

- Let $\mathcal{H}_1 = \{h \in \mathcal{H}_0 | h \text{ makes at least } \frac{\epsilon m}{4} \text{ mistakes in } R_1\}$. For any $h \in \mathcal{H}_1$, we let $X(h) \subset R_1$ denote a minimal set of examples in $R_1$ (breaking ties arbitrarily but in a fixed way) such that $h$ misclassify $X(h)$ and $|(\mathcal{G}X(h)) \cap S''_{\mathcal{X}}| \geq \frac{\epsilon m}{4}$ where $\mathcal{G}X(h) = \{gx | x \in X(h), g \in \mathcal{G}\}$ is the set of all examples lying in the orbits generated from $X(h)$. Let $K(h) = \mathcal{G}X(h)$ and $\mathcal{K} = \{K(h) | h \in \mathcal{H}_1\}$ the collection of all such sets. Notice that each example in $X(h)$ must belong to different orbits, otherwise it is not minimal. Besides, each orbit in $K(h)$ contains at least $\log^2 m$ examples from $S''_{\mathcal{X}}$ according to the definition of $R_1$. Hence, we have $|X(h)| \leq \frac{\epsilon m}{4 \log^2 m}$. Since there are at most $\frac{2m}{\log^2 m}$ orbits generated from $R_1$, we have $|\mathcal{K}| \leq \sum_{i=1}^{\frac{\epsilon m}{4\log^2 m}} \binom{\frac{2m}{\log^2 m}}{i} \leq \left(\frac{8e}{\epsilon}\right)^{\frac{\epsilon m}{4\log^2 m}}$. Recall that $\text{err}_{\mathcal{G}S}(h) = 0$ iff. $h(x) = y$ for all $(x, y) \in \mathcal{G}S$. Since $h$ misclassify $X(h)$, all examples in their orbits must go to $S'$ to guarantee $\text{err}_{\mathcal{G}S}(h) = 0$. Thus, we have

$$\Pr(\exists h \in \mathcal{H}_1, \text{err}_{S'}(h) \geq \frac{\epsilon}{2}, \text{err}_{\mathcal{G}S}(h) = 0)$$
$$\leq \Pr(\exists h \in \mathcal{H}_1, K(h) \cap S_{\mathcal{X}} = \emptyset) = \Pr(\exists K \in \mathcal{K}, K \cap S_{\mathcal{X}} = \emptyset)$$
$$\leq \sum_{K \in \mathcal{K}} 2^{-\frac{\epsilon m}{4}} \leq \left(\frac{8e}{\epsilon}\right)^{\frac{\epsilon m}{4\log^2 m}} \cdot 2^{-\frac{\epsilon m}{4}} = 2^{-\frac{\epsilon m}{4}\left(1 - \frac{\log(8e/\epsilon)}{\log^2 m}\right)} \leq 2^{-\frac{\epsilon m}{8}},$$

  when $m \geq \frac{8e}{\epsilon} + 4$.

- Let $\mathcal{H}_2 = \mathcal{H}_0 \setminus \mathcal{H}_1$. That is to say, for all $h \in \mathcal{H}_2$, $h$ will make at least $\frac{\epsilon m}{4}$ mistakes in $R_2$. Since $\text{VC}_{\text{ao}}(\mathcal{H}, \mathcal{G}) = d$ and every orbit generated from $R_2$ contains fewer than $\log^2 m$ examples in $S''_{\mathcal{X}}$, the number of examples in $R_2$ that can be shattered by $\mathcal{H}$ is no greater than $d \log^2 m$. Thus the number of ways labeling examples in $R_2$ is upper bounded by $\left(\frac{2em}{d}\right)^{d \log^2 m}$ by Sauer's lemma. Hence, we have

$$\Pr(\exists h \in \mathcal{H}_2, \text{err}_{S'}(h) \geq \frac{\epsilon}{2}, \text{err}_{\mathcal{G}S}(h) = 0) \leq \left(\frac{2em}{d}\right)^{d \log^2 m} \cdot 2^{-\frac{\epsilon m}{4}} = 2^{-\frac{\epsilon m}{4} + d \log^2 m \log(2em/d)}.$$

Combining the results for $\mathcal{H}_1$ and $\mathcal{H}_2$, we have

$$\Pr(B_{S,S'}) \leq \Pr(\exists h \in \mathcal{H}_1, \text{err}_{S'}(h) \geq \frac{\epsilon}{2}, \text{err}_{\mathcal{G}S}(h) = 0) + \Pr(\exists h \in \mathcal{H}_2, \text{err}_{S'}(h) \geq \frac{\epsilon}{2}, \text{err}_{\mathcal{G}S}(h) = 0)$$
$$\leq 2^{-\frac{\epsilon m}{8}} + 2^{-\frac{\epsilon m}{4} + d \log^2 m \log(2em/d)}$$
$$\leq \frac{\delta}{2},$$

when $m \geq \frac{8}{\epsilon}\left(d \log^2 m \log \frac{2em}{d} + \log \frac{4}{\delta} + e\right) + 4$. $\square$

## B  Proof of Theorem 2

*Proof.* For any $d > 0$, for any $\mathcal{X}, \mathcal{H}, \mathcal{G}$ satisfying that there exists a subset $X = \{x_0, x_1, \ldots, x_{2d}\} \subset \mathcal{X}$ such that

- their orbits are pairwise disjoint;

- for all $u \subset \{1, \ldots, 2d\}$ with $|u| = d$, there exists an $h_u \in \mathcal{H}$ such that $h_u(gx_i) = 1$ for all $g \in \mathcal{G}, i \in u \cup \{0\}$ and $h_u(x_i) = 0$ for $i \in \{0, \ldots, 2d\} \setminus u$,

we will prove the theorem for $\mathcal{H}' = \{h_u | u \subset \{1, \ldots, 2d\}, |u| = d\}$. For $\mathcal{X}, \mathcal{H}, \mathcal{G}$ satisfying the above conditions, we have $\mathrm{VC_{ao}}(\mathcal{H}', \mathcal{G}) \geq d$ and $\mathrm{VC_o}(\mathcal{H}', \mathcal{G})$ between 0 and $\mathrm{VC_{ao}}(\mathcal{H}', \mathcal{G})$. Then consider that the target function $h^* = h_{u^*}$ is chosen uniformly at random from $\mathcal{H}'$. The marginal data distribution $\mathcal{D}_{\mathcal{X}}$ puts probability mass $1 - 16\epsilon$ on $x_0$ and $\frac{16\epsilon}{d}$ on each point in $X_{u^*} := \{x_i | i \in u^*\}$. Then the target function $h^*$ is $(\mathcal{G}, \mathcal{D}_{\mathcal{X}})$-invariant.

Let the sample size $m = \frac{d}{64\epsilon}$. Given the training set $S_{\mathrm{trn}} \sim \mathcal{D}^m$, the expected number of sampled examples in $X_{u^*}$ is $\frac{d}{4}$. By Markov's inequality, with probability greater than $1/2$, we observed fewer than $\frac{d}{2}$ points of $X_{u^*}$ in $S_{\mathrm{trn}}$ (denoted as event $B$). Let $\mathcal{A}$ be any proper learner, which means $\mathcal{A}$ must output a hypothesis in $\mathcal{H}'$. For any $h \in \mathcal{H}'$ consistent with $\mathcal{G}S_{\mathrm{trn}}$, $h$ must predict $d$ unobserved points in $\{x_1, \ldots, x_{2d}\}$ as 0. Since for each unobserved point in $\{x_1, \ldots, x_{2d}\}$ labeled as 0 by $h$, conditioned on event $B$, this point has probability greater than $\frac{1}{3}$ to be in $X_{u^*}$, which implies it is misclassified by $h$. By following the stardard technique, let $\mathrm{err}'(h) = \mathrm{Pr}_{(x,y)\sim\mathcal{D}}(h(x) \neq y \wedge x \in X_{u^*})$, which is no greater than $\mathrm{err}(h)$ for any predictor $h$. Hence,

$$\mathbb{E}_{h^*, S_{\mathrm{trn}}}\left[\mathrm{err}'(\mathcal{A}(S_{\mathrm{trn}}))|B\right] = \mathbb{E}_{S_{\mathrm{trn}}}\left[\mathbb{E}_{h^*}\left[\mathrm{err}'(\mathcal{A}(S_{\mathrm{trn}}))|S_{\mathrm{trn}}, B\right]|B\right]$$

$$= \mathbb{E}_{S_{\mathrm{trn}}}\left[\mathbb{E}_{h^*}\left[\frac{16\epsilon}{d}\sum_{i\in[2d]:\mathcal{A}(S_{\mathrm{trn}},x_i)=0}\mathbb{1}[i \in u^*]|S_{\mathrm{trn}}, B\right]|B\right]$$

$$= \frac{16\epsilon}{d}\mathbb{E}_{S_{\mathrm{trn}}}\left[\sum_{i\in[2d]:\mathcal{A}(S_{\mathrm{trn}},i)=0}\mathbb{E}_{h^*}\left[\mathbb{1}[i \in u^*]|S_{\mathrm{trn}}, B\right]|B\right]$$

$$> \frac{16\epsilon}{d}\mathbb{E}_{S_{\mathrm{trn}}}\left[\sum_{i\in[2d]:\mathcal{A}(S_{\mathrm{trn}},i)=0}\frac{1}{3}|B\right]$$

$$= \frac{16\epsilon}{3} .$$

Then we have

$$\mathbb{E}_{h^*, S_{\mathrm{trn}}}\left[\mathrm{err}'(\mathcal{A}(S_{\mathrm{trn}}))\right] \geq \mathbb{E}_{h^*, S_{\mathrm{trn}}}\left[\mathrm{err}'(\mathcal{A}(S_{\mathrm{trn}}))|B\right] \cdot \mathrm{Pr}(B) > \frac{8\epsilon}{3} .$$

Thus, for any proper learner $\mathcal{A}$, there exists a target hypothesis $h^* \in \mathcal{H}'$ and a data distribution s.t. $\mathbb{E}_{S_{\mathrm{trn}}}[\mathrm{err}'(\mathcal{A}(S_{\mathrm{trn}}))] > 8\epsilon/3$. Since $\mathrm{err}'(h) \leq 16\epsilon$ for any predictor $h$, with probability greater than $\frac{1}{9}$, $\mathrm{err}(\mathcal{A}(S_{\mathrm{trn}})) \geq \mathrm{err}'(\mathcal{A}(S_{\mathrm{trn}})) > \epsilon$.

Here is an example of $\mathcal{X}, \mathcal{H}', \mathcal{G}$ satisfying the above conditions. Let $\mathcal{X} = \{0, \pm 1, \pm 2, \ldots, \pm 2d\}$. The group $\mathcal{G}$ is defined as $\mathcal{G}_d = \{e, -e\}$ where $e$ is the identity element. Thus $\mathcal{X}$ can be divided into $2d + 1$ pairwise disjoint orbits, $\{\{0\}, \{\pm 1\}, \ldots, \{\pm 2d\}\}$. For any $u \subset [2d]$ with $|u| = d$, define $h_u := 1 - \mathbb{1}[[2d] \setminus u]$, which labels $[2d] \setminus u$ by 0 and the other points by 1. Then we define the hypothesis class $\mathcal{H}' = \{h_u | |u| = d\}$. Since $\{1, \ldots, d\}$ can be shattered and no $d + 1$ points can be shattered by $\mathcal{H}'$, we have $\mathrm{VC_{ao}}(\mathcal{H}', \mathcal{G}) = d$. And since $\{0, -1, \ldots, -2d\}$ can only be labeled as 1 by $\mathcal{H}'$, we have $\mathrm{VC_o}(\mathcal{H}', \mathcal{G}) = 0$. $\qquad\square$

## C  Proof of Theorem 3

*Proof.* Let $d = \mathrm{VCdim}(\mathcal{H})$ and $X = \{x_1, x_2, \ldots, x_d\}$ be a set of examples shattered by $\mathcal{H}$. Let $\mathcal{S}_{d-1}$ denote the permutation group acting on $d - 1$ objects. Then for any partition $(A, [d-1] \setminus A)$

of $[d-1]$, we let $\mathcal{G}(A) := \{\sigma \in \mathcal{S}_{d-1} | \forall i \in A, \sigma(i) \in A\}$. By acting $\mathcal{G}(A)$ on $X$, then $X$ are partitioned into three orbits: $\{x_i | i \in A\}, \{x_i | i \notin A\}$ and $\{x_d\}$.

For convenience, we first consider the case of the instance space being $X$. Since there are only three orbits and every labeling of $X$ is realized by $\mathcal{H}$, $\mathrm{VC}_{\mathrm{ao}}(\mathcal{H}, \mathcal{G}(A)) = \mathrm{VC}_{\mathrm{o}}(\mathcal{H}, \mathcal{G}(A)) = 3$ for all $A \subset [d-1]$. Consider that we pick a set $A^*$ uniformly at random from $2^{[d-1]}$. Let $h^* = \mathbb{1}[\{x_i\}_{i \in A^*}]$ and $\mathcal{G} = \mathcal{G}(A^*)$. The data distribution put probability mass $1 - 16\epsilon$ on $x_d$ and the remaining $16\epsilon$ uniformly over $\{x_i | i \in [d-1]\}$. Then for the training set size $m = \frac{d-1}{64\epsilon}$, with probability at least $1/2$, at most half of $\{x_1, x_2, \ldots, x_{d-1}\}$ is sampled in the training set $S_{\mathrm{trn}}$. For any algorithm $\mathcal{A}$ not knowing $\mathcal{G}$, $\mathcal{A}$ will output a hypothesis $\widehat{h} = \mathcal{A}(S_{\mathrm{trn}})$, which does not depend on $\mathcal{G}$. For each unobserved point $x$ in $\{x_1, x_2, \ldots, x_{d-1}\}$, $\mathcal{A}$ has probability $1/2$ to misclassify $x$. Following the standard technique, let $\mathrm{err}'(h) := \mathrm{Pr}_{(x,y) \sim \mathcal{D}}(h(x) \neq y \wedge x \in \{x_1, x_2, \ldots, x_{d-1}\})$ and then we have

$$\mathbb{E}_{A^*, S_{\mathrm{trn}}}\left[\mathrm{err}'(\mathcal{A}(S_{\mathrm{trn}}))\right] \geq 2\epsilon,$$

which implies $\mathcal{M}_{\mathrm{INV}}(\epsilon, \delta; \mathcal{H}, \mathcal{G}, \mathcal{A}) = \Omega(\frac{d}{\epsilon} + \frac{1}{\epsilon}\log\frac{1}{\delta})$ by applying the standard technique in proving a lower bound of sample complexity in standard PAC learning.

In the case where the instance space not being $X$, we modify $\mathcal{G}(A^*)$ a little by arranging all points in $\{x | h^*(x) = 1\} \setminus X$ in one orbit and all points in $\{x | h^*(x) = 0\} \setminus X$ in another orbit. Then there are at most 5 orbits, and $\mathrm{VC}_{\mathrm{ao}}(\mathcal{H}, \mathcal{G}) = \mathrm{VC}_{\mathrm{o}}(\mathcal{H}, \mathcal{G}) \leq 5$. $\qquad \square$

# D Proof of Theorem 4

*Proof.* For any $k \leq \mathrm{VC}_{\mathrm{o}}(\mathcal{H}, \mathcal{G})$, let $X_k = \{x_1, \ldots, x_k\}$ be a set shattered in the way defined in Definition 4. Then $X_k$ can be shattered by $\mathcal{H}(X_k) = \{h_{|X_k} | h \in \mathcal{H} \text{ is } (\mathcal{G}, X_k)\text{-invariant}\}$ and $\mathrm{VCdim}(\mathcal{H}(X_k)) = k$. Since any data distribution $\mathcal{D}$ with $\mathcal{D}_{\mathcal{X}}(X_k) = 1$ is $\mathcal{G}$-invariant, any lower bound on the sample complexity of PAC learning of $\mathcal{H}(X_k)$ also lower bounds the sample complexity of invariantly realizable PAC learning of $\mathcal{H}$. Then the lower bound follows by standard arguments from Vapnik and Chervonenkis (1974); Blumer et al. (1989); Ehrenfeucht et al. (1989).

For the upper bound, recall that the algorithm $\mathcal{A}$ is defined by letting $\mathcal{A}(S)(x) = Q_{\mathcal{H}(X_S \cup \{x\}), X_S \cup \{x\}}(S, x)$ if $\mathcal{H}(X_S \cup \{x\}) \neq \emptyset$ and predicting arbitrarily if $\mathcal{H}(X_S \cup \{x\}) = \emptyset$ in Section 4.2. Due to the invariantly-realizable setting, if $S$ and the test point $(x, y)$ are i.i.d. from the data distribution, $h^*_{|X_S \cup \{x\}}$ is in $\mathcal{H}(X_S \cup \{x\})$ a.s. and then, $\mathcal{H}(X_S \cup \{x\})$ is nonempty. Following the analogous proof by Haussler et al. (1994) for standard PAC learning, we have

$$
\begin{aligned}
\mathbb{E}_{S \sim \mathcal{D}^t}\left[\mathrm{err}(\mathcal{A}(S))\right] &= \mathbb{E}_{(x_i, y_i)_{i \in [t+1]} \sim \mathcal{D}^{t+1}}\left[\mathbb{1}[\mathcal{A}(\{x_i, y_i\}_{i \in [t]}, x_{t+1}) \neq y_{t+1}]\right] \\
&= \frac{1}{(t+1)!}\sum_{\sigma \in \mathrm{Sym}(t+1)} \mathbb{E}\left[\mathbb{1}[\mathcal{A}(\{x_{\sigma(i)}, y_{\sigma(i)}\}_{i \in [t]}, x_{\sigma(t+1)}) \neq y_{\sigma(t+1)}]\right] \\
&= \mathbb{E}\left[\frac{1}{(t+1)!}\sum_{\sigma \in \mathrm{Sym}(t+1)} \mathbb{1}[\mathcal{A}(\{x_{\sigma(i)}, y_{\sigma(i)}\}_{i \in [t]}, x_{\sigma(t+1)}) \neq y_{\sigma(t+1)}]\right] \\
&\leq \frac{\mathbb{E}\left[\mathrm{VCdim}(\mathcal{H}(\{x_i | i \in [t+1]\}))\right]}{t+1} \qquad (6) \\
&\leq \frac{\mathrm{VC}_{\mathrm{o}}(\mathcal{H}, \mathcal{G})}{t+1}, \qquad (7)
\end{aligned}
$$

where Eq (6) adopts Lemma 1 and Eq (7) holds due to the definition of $\mathrm{VC}_{\mathrm{o}}(\mathcal{H}, \mathcal{G})$. To convert this algorithm, guaranteeing the expected error upper bounded by $\mathrm{VC}_{\mathrm{o}}(\mathcal{H}, \mathcal{G})$, into an algorithm with high probability $1 - \delta$, we again follow an argument of Haussler et al. (1994). Specifically, the algorithm runs $\mathcal{A}$ for $\lceil\log(2/\delta)\rceil$ times, each time using a new sample of size $\lceil 4\mathrm{VC}_{\mathrm{o}}(\mathcal{H}, \mathcal{G})/\epsilon\rceil$. Then the algorithm selects the hypothesis from the outputs with the minimal error on a new sample of size $\lceil 32/\epsilon(\ln(2/\delta) + \ln(\lceil\log(2/\delta)\rceil + 1))\rceil$. $\qquad \square$

# E   Proof of Theorem 5

We first introduce a useful lemma about a well-known Boosting algorithm, known as $\alpha$-Boost. Given access to a weak learning algorithm, it can output a hypothesis with strong learning guarantee. See Schapire and Freund (2012) for a proof.

**Lemma 2** (Boosting). *For any $k, n \in \mathbb{N}$ and multiset $(x_1, y_1), \ldots, (x_n, y_n) \in \mathcal{X} \times \mathcal{Y}$, suppose $\mathcal{A}_0$ is an algorithm that, for any distribution $\mathcal{P}$ on $\mathcal{X} \times \mathcal{Y}$ with $\mathcal{P}(\{(x_1, y_1), \ldots, (x_n, y_n)\}) = 1$, there exists $S_{\mathcal{P}} \in \{(x_1, y_1), \ldots, (x_n, y_n)\}^k$ with $\mathrm{err}_{\mathcal{P}}(\mathcal{A}_0(S_{\mathcal{P}})) \leq 1/3$. Then there is a numerical constant $c \geq 1$ such that, for $T = \lceil c \log(n) \rceil$, there exists multisets $S_1, \ldots, S_T \in \{(x_1, y_1), \ldots, (x_n, y_n)\}^k$ such that, for $\widehat{h}(\cdot) = \mathrm{Majority}(\mathcal{A}_0(S_1)(\cdot), \ldots, \mathcal{A}_0(S_T)(\cdot))$, it holds that $\widehat{h}(x_i) = y_i$ for all $i \in [n]$.*

Part of the proof relies on a well-known generalization bound for compression schemes. The following is the classic result due to Littlestone and Warmuth (1986).

**Lemma 3** (Consistent compression generalization bound). *There exists a finite numerical constant $c > 0$ such that, for any compression scheme $(\kappa, \rho)$, for any $n \in \mathbb{N}$ and $\delta \in (0, 1)$, for any distribution $\mathcal{D}$ on $\mathcal{X} \times \mathcal{Y}$, for $S \sim \mathcal{D}^n$, with probability at least $1 - \delta$, if $\mathrm{err}_S(\rho(\kappa(S))) = 0$, then*

$$\mathrm{err}_{\mathcal{D}}(\rho(\kappa(S))) \leq \frac{c}{n - |\kappa(S)|}(|\kappa(S)| \log(n) + \log(1/\delta)).$$

*Proof of the first part of Theorem 5.* The proof is inspired by the idea that representing algorithms by an orientation in a 1-inclusion graph in the transductive setting by Daniely and Shalev-Shwartz (2014). We will first prove a lower bound in the transductive setting and then extend the result to the inductive setting. For any $t \geq 2$, denote $\mu = \mu(\mathcal{H}, \mathcal{G}, t)$ and let $\phi$ with $|\phi| = t$ and $P \in \Delta(B(\mathcal{H}, \mathcal{G}, \phi))$ be given such that $\mu(\mathcal{H}, \mathcal{G}, \phi, P) \geq \frac{\mu}{2}$. A augmented dataset $\mathcal{G}S = \{(gx, y) | (x, y) \in S, g \in \mathcal{G}\}$ is the same as a multiset of labeled orbits $\{\mathcal{G}x \times \{y\} | (x, y) \in S\}$ up to different data formats. For convenience, we overload the notation a little by also allowing $\mathcal{A}$ being a mapping from a multiset of labeled orbits to a hypothesis. Then we construct a 1-inclusion graph $G_{\mathcal{H}, \mathcal{G}}(\phi) = \{V, E\}$ as introduced in Section 5 and define a mapping $w_{\mathcal{A}} \in W$ as follows. For any edge $e = \{\mathbf{f}, \mathbf{g}, x_i\} \in E$, let

$$w_{\mathcal{A}}(\{\mathbf{f}, \mathbf{g}, x_i\}, \mathbf{f}) = \Pr(\mathcal{A}(\phi_{-i}, \mathbf{f}_{-i}, x_i) = g_i),$$

and

$$w_{\mathcal{A}}(\{\mathbf{f}, \mathbf{g}, x_i\}, \mathbf{g}) = \Pr(\mathcal{A}(\phi_{-i}, \mathbf{f}_{-i}, x_i) = f_i),$$

which is well-defined as $\Pr(\mathcal{A}(\phi_{-i}, \mathbf{f}_{-i}, x_i) = g_i) + \Pr(\mathcal{A}(\phi_{-i}, \mathbf{f}_{-i}, x_i) = f_i) = 1$. Suppose our target function and the instance sequence $(\mathbf{f}, \mathbf{x})$ is drawn from the distribution $P$. Then the expected number of mistakes in the transductive learning setting is

$$\mathbb{E}_{(\mathbf{f}, \mathbf{x}) \sim P, \mathcal{A}} \left[ \sum_{i=1}^{t} \mathbb{1}[\mathcal{A}(\phi_{-i}, \mathbf{f}_{-i}, x_i) \neq f_i] \right]$$

$$\geq \mathbb{E}_{(\mathbf{f}, \mathbf{x}) \sim P} \left[ \sum_{i \in [t]: \exists e \in E, \{\mathbf{f}, x_i\} \subset e} w_{\mathcal{A}}(e(\mathbf{f}, x_i), \mathbf{f}) \right]$$

$$\geq \min_{w} \mathbb{E}_{(\mathbf{f}, \mathbf{x}) \sim P} \left[ \sum_{i \in [t]: \exists e \in E, \{\mathbf{f}, x_i\} \subset e} w(e(\mathbf{f}, x_i), \mathbf{f}) \right]$$

$$= \mu(\mathcal{H}, \mathcal{G}, \phi, P), \tag{8}$$

where Eq (8) holds due to the definition of $\mu(\mathcal{H}, \mathcal{G}, \phi, P)$.

Now we prove the lower bound in the inductive setting based on the similar idea. Let $\phi$ and $P$ be the same as those in the transductive setting. For any $\epsilon < \frac{\mu}{16(t-1)}$, we draw $(\mathbf{f}, \mathbf{x}) \sim P$ and then let $\mathbf{f}$ be our target function and let the marginal data distribution be like, putting probability mass $1 - \frac{16(t-1)\epsilon}{\mu}$ on $x_t$ and the remaining probability mass uniformly over $\{x_1, \ldots, x_{t-1}\}$. Denote this data distribution by $\mathcal{D}_{\mathbf{f}, \mathbf{x}}$. For any fixed $i \in [t-1]$, $\Pr(x_i \notin S_{\mathrm{trn}, \mathcal{X}}) = (1 - \frac{16\epsilon}{\mu})^m \geq (\frac{1}{4})^{\frac{16m\epsilon}{\mu}} \geq \frac{1}{2}$

when $m \leq \frac{\mu}{32\epsilon}$. For any hypothesis $h$, let $\mathrm{err}'_{\mathcal{D}}(h) = \Pr_{(x,y)\sim\mathcal{D}}(h(x) \neq y$ and $x \in \{x_i\}_{i\in[t-1]})$ and we always have $\mathrm{err}(h) \geq \mathrm{err}'(h)$. Then we have

$$\mathbb{E}_{(\mathbf{f},\mathbf{x})\sim P, S_{\mathrm{trn}}\sim\mathcal{D}^m_{\mathbf{f},\mathbf{x}}, \mathcal{A}}\left[\mathrm{err}'(\mathcal{A}(\mathcal{G}S_{\mathrm{trn}}))\right]$$

$$=\frac{16\epsilon}{\mu}\sum_{i=1}^{t-1}\mathbb{E}_{(\mathbf{f},\mathbf{x})\sim P, S_{\mathrm{trn}}\sim\mathcal{D}^m_{\mathbf{f},\mathbf{x}}, \mathcal{A}}\left[\mathbb{1}[\mathcal{A}(\mathcal{G}S_{\mathrm{trn}}, x_i) \neq f_i]\right]$$

$$\geq\frac{16\epsilon}{\mu}\sum_{i=1}^{t-1}\mathbb{E}_{(\mathbf{f},\mathbf{x})\sim P}\left[\Pr_{S_{\mathrm{trn}}\sim\mathcal{D}^m_{\mathbf{f},\mathbf{x}}, \mathcal{A}}(\mathcal{A}(\mathcal{G}S_{\mathrm{trn}}, x_i) \neq f_i|x_i \notin S_{\mathrm{trn},\mathcal{X}})\Pr(x_i \notin S_{\mathrm{trn},\mathcal{X}})\right]$$

$$\geq\frac{8\epsilon}{\mu}\mathbb{E}_{(\mathbf{f},\mathbf{x})\sim P}\left[\sum_{i=1}^{t-1}\Pr_{S_{\mathrm{trn}}\sim\mathcal{D}^m_{\mathbf{f},\mathbf{x}}, \mathcal{A}}(\mathcal{A}(\mathcal{G}S_{\mathrm{trn}}, x_i) \neq f_i|x_i \notin S_{\mathrm{trn},\mathcal{X}})\right]. \tag{9}$$

For all $i \in [t-1]$, for all $z \in \phi_i$, if there is an edge $e = \{\mathbf{f}, \mathbf{g}, z\} \in E$, we let

$$\widetilde{w}(e, \mathbf{f}) = \Pr_{S_{\mathrm{trn}}\sim\mathcal{D}^m_{\mathbf{f},\mathbf{x}}, \mathcal{A}}(\mathcal{A}(\mathcal{G}S_{\mathrm{trn}}, z) \neq f_i|z \notin S_{\mathrm{trn},\mathcal{X}}),$$

where $\mathbf{x}$ is an arbitrary sequence in $\mathcal{U}_{\mathbf{f}}(\phi)$ satisfying that $x_i = z$. Then $\widetilde{w}(e, \mathbf{f})$ is well-defined since the distribution of $\mathcal{G}S_{\mathrm{trn}}$ conditioned on $x_i \notin S_{\mathrm{trn},\mathcal{X}}$ is the same for all $\mathbf{x} \in \mathcal{U}_{\mathbf{f}}(\phi)$ with $x_i = z$. Actually, conditioned on $x_i \notin S_{\mathrm{trn},\mathcal{X}}$, the distribution of $\mathcal{G}S_{\mathrm{trn}}$ is also the same when $S_{\mathrm{trn}}$ is sampled from $\mathcal{D}_{\mathbf{g},\mathbf{x}'}$ where $\mathbf{x}'$ is an arbitrary sequence in $\mathcal{U}_{\mathbf{g}}(\phi)$ satisfying that $x'_i = z$. Hence, $\widetilde{w}(e, \mathbf{f}) + \widetilde{w}(e, \mathbf{g}) = 1$. By letting $\widetilde{w}(e, \mathbf{h}) = 0$ for all $\mathbf{h} \notin e$, $\widetilde{w}$ is in $W$. Then we have

$$\text{Eq (9)} \geq \frac{8\epsilon}{\mu}\mathbb{E}_{(\mathbf{f},\mathbf{x})\sim P}\left[\sum_{i\in[t-1]:\exists e\in E, \{\mathbf{f}, x_i\}\subset e}\widetilde{w}(e, \mathbf{f})\right]$$

$$\geq \frac{8\epsilon}{\mu}\min_{w\in W}\mathbb{E}_{(\mathbf{f},\mathbf{x})\sim P}\left[\sum_{i\in[t]:\exists e\in E, \{\mathbf{f}, x_i\}\subset e}w(e, \mathbf{f}) - 1\right]$$

$$\geq 4\epsilon(1 - 2/\mu) \geq 2\epsilon,$$

when $\mu \geq 4$. Hence, there exists a labeling function $\mathbf{f}$ (i.e., there exists a target function $h^*$) and a data distribution $\mathcal{D}_{\mathbf{f},\mathbf{x}}$ such that $\mathbb{E}_{S_{\mathrm{trn}}, \mathcal{A}}\left[\mathrm{err}'(\mathcal{A}(\mathcal{G}S_{\mathrm{trn}}))\right] \geq 2\epsilon$. Since $\mathrm{err}'(h) \leq \frac{16(t-1)\epsilon}{\mu}$ for all hypothesis $h$, $\Pr(\mathrm{err}(\mathcal{A}(\mathcal{G}S_{\mathrm{trn}})) > \epsilon) \geq \Pr(\mathrm{err}'(\mathcal{A}(\mathcal{G}S_{\mathrm{trn}})) > \epsilon) > \frac{\mu}{16(t-1)}$. $\qquad\square$

*Proof of the second and third parts of Theorem 5.* For any $n \in \mathbb{N}$, for any given sample $\{(x_1, y_1), \ldots, (x_{n+1}, y_{n+1})\}$, let $\phi = \{\phi_1, \ldots, \phi_{n+1}\} = \{\mathcal{G}x_1, \ldots, \mathcal{G}x_{n+1}\}$ denote the multi-set of $t + 1$ orbits and construct the one-inclusion graph $G_{\mathcal{H},\mathcal{G}}(\phi)$. As mentioned in Definition 6, every $w \in W$ defines a randomized orientation of each edge in graph $G_{\mathcal{H},\mathcal{G}}(\phi)$. That is, for any fixed $w \in W$, for every edge $e = \{\mathbf{f}, \mathbf{g}, x\}$, $w$ defines a probability over $\{\mathbf{f}, \mathbf{g}\}$. Then we can construct an algorithm $\mathcal{A}_w$ for each $w \in W$.

Given the input $(\phi_{-i}, \mathbf{y}_{-i}, x_i)$, $\mathcal{A}_w$ finds the subset of vertices $\{(\mathbf{f}, x_i)|\mathbf{f}_{-i} = \mathbf{y}_{-i}\}$ whose labelings are consistent with $(\phi_{-i}, \mathbf{y}_{-i})$. If there exist two such vertices, $(\mathbf{f}, x_i)$ and $(\mathbf{g}, x_i)$, they must be connected by $e = (\mathbf{f}, \mathbf{g}, x_i)$ due to the definition of $G_{\mathcal{H},\mathcal{G}}(\phi)$. Then $\mathcal{A}_w$ will predict $x_i$ as $f_i$ with probability $w(e, \mathbf{g})$ and as $g_i$ with probability $w(e, \mathbf{f})$. If only one such vertex $(\mathbf{f}, x_i)$ exists, $\mathcal{A}_w$ predicts the label of $x_i$ by $f_i$. Due to the realizable setting, there must exist at least one such vertex and the algorithm's prediction must be correct when only one vertex exists.

To complete the algorithm, the remaining part is how to choose a good $w \in W$. For any true labeling $\mathbf{f}^*$ and any sequence of natural data $\mathbf{x}^* \in \mathcal{U}_{\mathbf{f}^*}(\phi)$, for each $i \in [n + 1]$, if there is an edge $e \supset \{\mathbf{f}^*, x_i^*\}$, it means the algorithm possibly misclassify $x_i^*$ (with the probability dependent on $w$); if there is no such edge, it means the algorithm will not misclassify $x_i^*$ no matter what $w$ is. For any labeling $\mathbf{f} \in \Pi_{\mathcal{H}}(\phi)$ and a sequence of natural data $\mathbf{x} \in \mathcal{U}_f(\phi)$, we can represent the subset of the points in $\mathbf{x}$ that the algorithm is uncertain about by a mapping $a_{\mathbf{f},\mathbf{x}} : E \times \Pi_{\mathcal{H}}(\phi) \mapsto \{0, 1\}$ where $a_{\mathbf{f},\mathbf{x}}(e, \mathbf{g}) = 1$ iff. $\mathbf{g} = \mathbf{f}$ and there exists $i \in [n + 1]$ s.t. $\{\mathbf{f}, x_i\} \in e$. Due to the definition, $a_{\mathbf{f},\mathbf{x}}$ has at most $n + 1$ non-zero entries. Let $A = \{a_{\mathbf{f},\mathbf{x}}|\mathbf{f} \in \Pi_{\mathcal{H}}(\phi), \mathbf{x} \in \mathcal{U}_f(\phi)\}$ denote the set of

all such mappings. We now first consider the case where $|E| < \infty$. Then for a training set of size $n$, we can rewrite the expected error as

$$\mathbb{E}_{S \sim \mathcal{D}^n} \left[ \mathrm{err}(\mathcal{A}_w(\mathcal{G}S)) \right]$$

$$= \mathbb{E}_{S \sim \mathcal{D}^n, (x,y) \sim \mathcal{D}, \mathcal{A}_w} \left[ \mathbb{1}[\mathcal{A}_w(\mathcal{G}S, x) \neq y] \right]$$

$$= \mathbb{E}_{(\mathbf{x},\mathbf{y}) \sim \mathcal{D}^{n+1}, \mathcal{A}_w} \left[ \frac{1}{n+1} \sum_{i=1}^{n+1} \mathbb{1}[\mathcal{A}_w(\boldsymbol{\phi}_{-i}, \mathbf{y}_{-i}, x_i) \neq y_i] \right]$$

$$= \mathbb{E}_{(\mathbf{x},\mathbf{y}) \sim \mathcal{D}^{n+1}, \mathcal{A}_w} \left[ \frac{1}{n+1} \sum_{i=1}^{n+1} \mathbb{1}[\mathcal{A}_w(\boldsymbol{\phi}_{-i}, \mathbf{y}_{-i}, x_i) \neq y_i] \mathbb{1}[\exists e \supset \{x_i, \mathbf{f}^*\}] \right]$$

$$= \mathbb{E}_{(\mathbf{x},\mathbf{y}) \sim \mathcal{D}^{n+1}} \left[ \frac{1}{n+1} \sum_{e \in E} w(e, \mathbf{f}^*) a_{\mathbf{f}^*, \mathbf{x}}(e, \mathbf{f}^*) \right]$$

$$= \mathbb{E}_{(\mathbf{x},\mathbf{y}) \sim \mathcal{D}^{n+1}} \left[ \frac{1}{n+1} \sum_{e \in E, \mathbf{f} \in \Pi_{\mathcal{H}}(\boldsymbol{\phi})} w(e, \mathbf{f}) a_{\mathbf{f}^*, \mathbf{x}}(e, \mathbf{f}) \right] . \tag{10}$$

where the last equality holds due to $a_{\mathbf{f}^*, \mathbf{x}}(e, \mathbf{f}) = 0$ for all $\mathbf{f} \neq \mathbf{f}^*$. Since $\mathbf{f}^*$ and $\mathbf{x}$ is unknown, our goal is to find a $w$ with $\sum_{e \in E, \mathbf{f} \in \Pi_{\mathcal{H}}(\boldsymbol{\phi})} w(e, \mathbf{f}) a_{\mathbf{f}^*, \mathbf{x}}(e, \mathbf{f})$ upper bounded for all $\mathbf{f}^*$ and $\mathbf{x}$. The algorithm picks $w^* = \arg \min_{w \in W} \max_{a \in A} \sum_{e \in E, \mathbf{f} \in \Pi_{\mathcal{H}}(\boldsymbol{\phi})} w(e, \mathbf{f}) a(e, \mathbf{f})$. Then we have

$$\max_{a \in A} \sum_{e \in E, \mathbf{f} \in \Pi_{\mathcal{H}}(\boldsymbol{\phi})} w^*(e, \mathbf{f}) a(e, \mathbf{f}) = \min_{w \in W} \max_{a \in A} \sum_{e \in E, \mathbf{f} \in \Pi_{\mathcal{H}}(\boldsymbol{\phi})} w(e, \mathbf{f}) a(e, \mathbf{f})$$

$$= \min_{w \in W} \max_{a \in \mathrm{Conv}(A)} \sum_{e \in E, \mathbf{f} \in \Pi_{\mathcal{H}}(\boldsymbol{\phi})} w(e, \mathbf{f}) a(e, \mathbf{f}) = \max_{a \in \mathrm{Conv}(A)} \min_{w \in W} \sum_{e \in E, \mathbf{f} \in \Pi_{\mathcal{H}}(\boldsymbol{\phi})} w(e, \mathbf{f}) a(e, \mathbf{f}) , \tag{11}$$

where the last equality is due to Minimax theorem. Since the optimal solution $a^*$ to Eq. (11) is in the convex hull of $A$, there is a distribution $P^* \in \Delta(B(\mathcal{H}, \mathcal{G}, \boldsymbol{\phi}))$ such that $a^* = \mathbb{E}_{(\mathbf{f}^*, \mathbf{x}) \sim P^*} [a_{\mathbf{f}^*, \mathbf{x}}]$. Then we have

$$\text{Eq (11)} = \min_{w \in W} \sum_{e \in E, \mathbf{f} \in \Pi_{\mathcal{H}}(\boldsymbol{\phi})} w(e, \mathbf{f}) \mathbb{E}_{(\mathbf{f}^*, \mathbf{x}) \sim P^*} [a_{\mathbf{f}^*, \mathbf{x}}(e, \mathbf{f})]$$

$$= \min_{w \in W} \mathbb{E}_{(\mathbf{f}^*, \mathbf{x}) \sim P^*} \left[ \sum_{e \in E, \mathbf{f} \in \Pi_{\mathcal{H}}(\boldsymbol{\phi})} w(e, \mathbf{f}) a_{\mathbf{f}^*, \mathbf{x}}(e, \mathbf{f}) \right]$$

$$= \min_{w \in W} \mathbb{E}_{(\mathbf{f}^*, \mathbf{x}) \sim P^*} \left[ \sum_{i \in [n+1] : \exists e \in E, \{\mathbf{f}^*, x_i\} \subset e} w(e(\mathbf{f}^*, x_i), \mathbf{f}^*) \right]$$

$$= \mu(\mathcal{H}, \mathcal{G}, \boldsymbol{\phi}, P^*) .$$

Combined with Eq (10), we have $\mathbb{E}_{S \sim \mathcal{D}^n} \left[ \mathrm{err}(\mathcal{A}_{w^*}(\mathcal{G}S)) \right] \leq \frac{\mu(\mathcal{H}, \mathcal{G}, \boldsymbol{\phi}, n+1)}{n+1}$.

For $|E| = \infty$, $E \times \Pi_{\mathcal{H}}(\boldsymbol{\phi})$ could be infinite dimensional and thus, we need to use Sion's minimax theorem. The details of how to apply Sion's minimax theorem are described as follows. For all $w \in W$, for any edge $e = \{\mathbf{f}, \mathbf{g}, x\}$, we have $w(e, \mathbf{f}) + w(e, \mathbf{g}) = 1$ and thus, there exists a one-to-one mapping $\beta : W \mapsto [0, 1]^E$ where $\beta(w)(e) = w(e, \mathbf{f}_{e,0})$ where $\mathbf{f}_{e,0}$ is the labeling in $e$ predicting $x$ as zero. In the following, we will overload the notation by using $w$ to represent $\beta(w)$ when it is clear from the context that it is in the space $[0, 1]^E$. Then we define a mapping $\mathrm{BL}(\cdot, \cdot) : [0, 1]^E \times A$ by

$$\mathrm{BL}(w, a) := \sum_{e : a(e, \mathbf{f}_{e,0}) = 1} w(e) + \sum_{e : a(e, \mathbf{f}_{e,1}) = 1} (1 - w(e)),$$

where $\mathbf{f}_{e,0}, \mathbf{f}_{e,1}$ are labelings in $e = \{\mathbf{f}_{e,0}, \mathbf{f}_{e,1}, x\}$ and they label $x$ as 0 and 1 respectively. Then similar to Eq (10), we can represent the expected error as

$$\mathbb{E}_{S \sim \mathcal{D}^n} \left[ \mathrm{err}(\mathcal{A}_w(\mathcal{G}S)) \right] = \mathbb{E}_{(\mathbf{x},\mathbf{y}) \sim \mathcal{D}^{n+1}} \left[ \frac{1}{n+1} \mathrm{BL}(w, a_{\mathbf{f}^*, \mathbf{x}}) \right] .$$

Now we want to upper bound $\min_{w \in [0,1]^E} \sup_{a \in A} \mathrm{BL}(w, a)$. $\mathrm{BL}(\cdot, \cdot)$ can be extended to $\mathbb{R}^E \times A$ by letting $\mathrm{BL}(w, a) = \sum_{e:a(e, \mathbf{f}_{e,0})=1} w(e) + \sum_{e:a(e, \mathbf{f}_{e,1})=1}(1 - w(e))$ for $w \in \mathbb{R}^E$. Since $a$ has at most $n + 1$ non-zeros entries, $|\mathrm{BL}(w, a)| \leq (n + 1) \max(\|w\|_\infty, \|\mathbf{1} - w\|_\infty)$. Let $\widetilde{A} = \{\sum_{i=1}^N c_i a_i | \forall i, a_i \in A, c_i > 0, \sum_{i=1}^N c_i = 1, N \in \mathbb{N}\}$ be the set of all finite convex combination of elements in $A$. We extend BL from $R^E \times A$ to $R^E \times \widetilde{A}$ by defining $\mathrm{BL}(w, a) = \sum_{i=1}^N c_i \mathrm{BL}(w, a_i)$ for $a = \sum_{i=1}^N c_i a_i \in \widetilde{A}$.

We define a metric $d$ in $\widetilde{A}$: for $a = \sum_{i=1}^N c(a_i) a_i$ and $a' = \sum_{i=1}^{N'} c'(a_i') a_i'$, the distance between $a$ and $a'$ is $d(a, a') := \sum_{\alpha \in A: c(\alpha) \neq 0 \text{ or } c'(\alpha) \neq 0} |c(\alpha) - c'(\alpha)|$. For any fixed $w \in \mathbb{R}^E$, for any $a \in \widetilde{A}$, for every open ball $\mathcal{B}_r(\mathrm{BL}(w, a))$ centered at $\mathrm{BL}(w, a)$ with radius $r > 0$, there is an open ball $\mathcal{B}_{r'}(a)$ in the metric space $(\widetilde{A}, d)$ with $r' = \frac{r}{(n+1)(\max(\|w\|_\infty, \|\mathbf{1}-w\|_\infty))}$ such that for all $a' \in \mathcal{B}_{r'}(a)$, $|\mathrm{BL}(w, a') - \mathrm{BL}(w, a)| \leq \sum_{\alpha \in \{a_i\}_{i \in [N]} \cup \{a_i'\}_{i \in [N']}} |c(\alpha) - c'(\alpha)| |\mathrm{BL}(w, \alpha)| < r' \cdot |\mathrm{BL}(w, \alpha)| \leq r$. Hence, $\mathrm{BL}(w, \cdot)$ is continuous for all $w \in \mathbb{R}^E$.

Consider the the standard topology in $\mathbb{R}$ and then $[0, 1]$ is compact. Then let $\mathcal{T}$ be the product topology of $\mathbb{R}^E$. Then by Tychonoff theorem, $[0, 1]^E$ is compact in $\mathbb{R}^E$. For any fixed $a = \sum_{i=1}^N c_i a_i \in \widetilde{A}$, there are at most $N(n+1)$ non-zero entries. Then for any $w \in \mathbb{R}^E$, for every open ball $\mathcal{B}_r(\mathrm{BL}(w, a))$ centered at $\mathrm{BL}(w, a)$ with radius $r > 0$, then there is a neighborhood $U = \prod_{e \in E} S_e$, where $S_e = (w_e - \frac{r}{N(n+1)}, w_e + \frac{r}{N(n+1)})$ if at least one of $a(e, \mathbf{f}_{e,0}), a(e, \mathbf{f}_{e,1})$ is non-zero and $S_e = \mathbb{R}$ for other $e \in E$, such that $|\mathrm{BL}(w', a) - \mathrm{BL}(w, a)| < r$ for all $w' \in U$. That is, $\mathrm{BL}(U) \subset \mathcal{B}_r(\mathrm{BL}(w, a))$. Hence, $\mathrm{BL}(\cdot, a)$ is continuous for all $a \in \widetilde{A}$.

It is easy to check that $\mathrm{BL}(\cdot, a)$ and $\mathrm{BL}(w, \cdot)$ are linear for all $a \in \widetilde{A}$, $w \in W$. Then by Sion's minimax theorem, we have

$$\min_{w \in [0,1]^E} \sup_{a \in A} \mathrm{BL}(w, a) \leq \min_{w \in [0,1]^E} \sup_{a \in \widetilde{A}} \mathrm{BL}(w, a) = \sup_{a \in \widetilde{A}} \min_{w \in [0,1]^E} \mathrm{BL}(w, a) =: v^* .$$

Let $w^* = \arg\min_{w \in [0,1]^E} \sup_{a \in A} \mathrm{BL}(w, a)$. There exists a sequence $a_1, a_2, \ldots$ in $\widetilde{A}$ such that $\lim_{k \to \infty} \min_{w \in [0,1]^E} \mathrm{BL}(w, a_k) = v^*$. For each $a_k$, we let $P_k \in \Delta(B(\mathcal{H}, \mathcal{G}, \phi))$ be the distribution over $B(\mathcal{H}, \mathcal{G}, \phi)$ such that $a_k = \mathbb{E}_{(\mathbf{f}^*, \mathbf{x}) \sim P_k}[a_{\mathbf{f}^*, \mathbf{x}}]$. Due to the definition of $\widetilde{A}$, we know that $P_k$ is a discrete distribution with finite support. Then we have

$$\begin{aligned}
\sup_{a \in A} \mathrm{BL}(w^*, a) &\leq \min_{w \in [0,1]^E} \sup_{a \in \widetilde{A}} \mathrm{BL}(w, a) \\
&= \sup_{a \in \widetilde{A}} \min_{w \in [0,1]^E} \mathrm{BL}(w, a) \\
&= \lim_{k \to \infty} \min_{w \in [0,1]^E} \mathrm{BL}(w, \mathbb{E}_{(\mathbf{f}^*, \mathbf{x}) \sim P_k}[a_{\mathbf{f}^*, \mathbf{x}}]) \\
&= \lim_{k \to \infty} \min_{w \in [0,1]^E} \mathbb{E}_{(\mathbf{f}^*, \mathbf{x}) \sim P_k}[\mathrm{BL}(w, a_{\mathbf{f}^*, \mathbf{x}})] \\
&\leq \sup_{P \in \Delta(B(\mathcal{H}, \mathcal{G}, \phi))} \min_{w \in [0,1]^E} \mathbb{E}_{(\mathbf{f}^*, \mathbf{x}) \sim P}[\mathrm{BL}(w, a_{\mathbf{f}^*, \mathbf{x}})] \\
&= \mu(\mathcal{H}, \mathcal{G}, \phi, n + 1) .
\end{aligned}$$

Hence, $\mathbb{E}_{S \sim \mathcal{D}^n}[\mathrm{err}(\mathcal{A}_{w^*}(\mathcal{G}S))] \leq \frac{\mu(\mathcal{H}, \mathcal{G}, \phi, n+1)}{n+1}$.

**The first upper bound**  We can convert the above bound into a high probability bound by $\alpha$-Boost. Let $\mathcal{A}_0 = \mathcal{A}_{w^*}$ as defined above. Let $t$ be any positive integer such that $\frac{1}{6} \leq \frac{\mu(\mathcal{H}, \mathcal{G}, t)}{t} \leq \frac{1}{3}$. As established above, for $S_{\mathrm{trn}} = \{(x_1, y_1), \ldots, (x_m, y_m)\} \sim \mathcal{D}^m$ and any distribution $\mathcal{P}$ supported on $\{(x_1, y_1), \ldots, (x_m, y_m)\}$, for $k = t - 1$ and $S \sim \mathcal{P}^k$, $\mathbb{E}[\mathrm{err}_{\mathcal{P}}(\mathcal{A}_0(S))] \leq 1/3$. Thus given $S_{\mathrm{trn}}$ and $\mathcal{P}$, there exists a deterministic choice of $S_{\mathcal{P}} \in \{(x_1, y_1), \ldots, (x_m, y_m)\}^k$ with $\mathrm{err}_{\mathcal{P}}(\mathcal{A}_0(S_{\mathcal{P}})) \leq 1/3$. Then Lemma 2 implies that for a value $T = \lceil c_1 \log m \rceil$ (for numerical constant $c_1 \geq 1$), there exists $S_1, \ldots, S_T \in \{(x_1, y_1), \ldots, (x_m, y_m)\}^k$ such that, for $\widehat{h}(\cdot) = \mathrm{Majority}(\mathcal{A}_0(S_1)(\cdot), \ldots, \mathcal{A}_0(S_T)(\cdot))$, it holds that $\widehat{h}(x_i) = y_i$ for all $i \in [m]$. Note that $\widehat{h}$

can be expressed as a compression scheme. By Lemma 3, with probability at least $1 - \delta$,

$$\mathrm{err}(\widehat{h}) \leq \frac{c_2}{m - kT} \left( kT \log m + \log \frac{1}{\delta} \right) ,$$

for a numerical constant $c_2 \geq 1$. Thus, for any given $\epsilon \in (0,1)$, the right hand side can be made less than $\epsilon$ for an appropriate choice of

$$m = O(\frac{1}{\epsilon}(k \log^2 \frac{k}{\epsilon} + \log \frac{1}{\delta})) ,$$

where $k = t - 1 \leq 6\mu(\mathcal{H}, \mathcal{G}, t) - 1$.

**The second upper bound**    Again, using the same standard technique as we used in Theorem 4 to convert an algorithm with expected error upper bound to an algorithm with high probability guarantee. The algorithm runs $\mathcal{A}_{w^*}$ for $\lceil \log(2/\delta) \rceil$ times, each time using a new sample of size $\lceil 4\mu(\mathcal{H}, \mathcal{G})/\epsilon \rceil$. Then the algorithm selects the hypothesis from the outputs with the minimal error on a new sample of size $\lceil 32/\epsilon(\ln(2/\delta) + \ln(\lceil \log(2/\delta) \rceil + 1)) \rceil$. □

# F    Proof of Theorem 6

*Proof.* Let $d = \dim(\mathcal{H}, \mathcal{G})$. Let $\phi$ and the corresponding $B = \{(\mathbf{f}, \mathbf{x_f})\}_{\mathbf{f} \in \mathcal{Y}^d}$ be given. Let $P$ be the uniform distribution over $B$. Then

$$\mu(\mathcal{H}, \mathcal{G}, \phi, P) = \min_{w \in W} \mathbb{E}_{(\mathbf{f}, \mathbf{x_f}) \sim P} \left[ \sum_{i \in [d]: \exists e \in E, \{\mathbf{f}, x_{\mathbf{f},i}\} \subset e} w(e, \mathbf{f}) \right]$$

$$= \min_{w} \frac{1}{2^d} \sum_{\mathbf{f} \in \mathcal{Y}^d} \sum_{i \in [d]: \exists e \in E, \{\mathbf{f}, x_{\mathbf{f},i}\} \subset e} w(e, \mathbf{f})$$

$$= \frac{1}{2^d} \min_{w} \sum_{\mathbf{f} \in \mathcal{Y}^d} \sum_{i=1}^{d} \frac{1}{2} \left( w(e(\mathbf{f}, x_{\mathbf{f},i}), \mathbf{f}) + w(e(\mathbf{f}, x_{\mathbf{f},i}), (1 - f_i, \mathbf{f}_{-i})) \right) \qquad (12)$$

$$= \frac{1}{2^d} \cdot \frac{2^d \cdot d}{2} = \frac{d}{2} ,$$

where Eq (12) holds since $x_{\mathbf{f},i} = x_{\mathbf{g},i}$ if $\mathbf{f} \oplus \mathbf{g} = \mathbf{e}_i$ due to the definition of $\dim(\mathcal{H}, \mathcal{G})$. □

# G    Proof of Theorem 7

*Proof of the lower bound.* Let $d = \mathrm{VC}_{\mathrm{ao}}(\mathcal{H}, \mathcal{G})$, let $X_d = \{x_1, \ldots, x_d\}$ be a set shattered in the way defined in Definition 5 and let $\mathcal{H}' := \{h_{|X_d} | h \in \mathcal{H}\}$. Then $\mathrm{VCdim}(\mathcal{H}') = d$ and any lower bound on the sample complexity of PAC learning of $\mathcal{H}'$ is also a lower bound on the sample complexity of relaxed realizable PAC learning of $\mathcal{H}$. The lower bound follow by standard arguments from Vapnik and Chervonenkis (1974); Blumer et al. (1989); Ehrenfeucht et al. (1989). □

*Proof of the upper bound of the algorithm ERM-INV.* The algorithm ERM-INV works as follows. ERM-INV first applies ERM over the original data set. Then for every test instance $x$, if $x$ lies in the orbits generated by the original data, the algorithm predicts $x$ by the label of the training instance in the same orbit; otherwise, the algorithm predicts according to the ERM output. Specifically, given the training set $S_{\mathrm{trn}} = \{(x_1, y_1), \ldots, (x_m, y_m)\} \sim \mathcal{D}^m$, the algorithm finds a hypothesis $h \in \mathcal{H}$ consistent with $S_{\mathrm{trn}}$ and then outputs $f_{h, S_{\mathrm{trn}}}$ defined by

$$f_{h, S_{\mathrm{trn}}}(x) = \begin{cases} y_i & \text{if there exists } i \in [m] \text{ s.t. } x \in \mathcal{G}x_i , \\ h(x) & \text{o.w.} \end{cases} \qquad (13)$$

The function $f_{h, S_{\mathrm{trn}}}$ is well-defined a.s. when the data distribution $\mathcal{D}$ is $\mathcal{G}$-invariant since if there exists $i \neq j$ such that $x_i \in \mathcal{G}x_j$, then $y_i = y_j = h^*(x_j)$.

The proof idea is similar to that of Theorem 1. Let $d = \text{VC}_{\text{ao}}(\mathcal{H}, \mathcal{G})$. Consider two sets $S$ and $S'$ of $m$ i.i.d. samples drawn from the data distribution $\mathcal{D}$ each. We denote $A_S$ the event of $\{\exists h \in \mathcal{H}, \text{err}_S(h) = 0, \text{err}_{\mathcal{D}}(f_{h,S}) \geq \epsilon\}$ and $B_{S,S'} = \{\exists h \in \mathcal{H}, \text{err}_S(h) = 0, \text{err}_{S'}(f_{h,S}) \geq \frac{\epsilon}{2}\}$. By Chernoff bound, we have $\Pr(B_{S,S'}) \geq \Pr(A_S) \cdot \Pr(B_{S,S'}|A_S) \geq \frac{1}{2}\Pr(A_S)$ when $m \geq \frac{8}{\epsilon}$. The sampling process of $S$ and $S'$ is equivalent to drawing $2m$ i.i.d. samples and then randomly partitioning into $S$ and $S'$ of $m$ each. For any fixed $S''$, let us divide $S''$ into two categories in terms of the number of examples in each orbit. Let $R_1 = \{x \mid |\mathcal{G}x \cap S''_{\mathcal{X}}| \geq \log^2 m\}$ and $R_2 = S''_{\mathcal{X}} \setminus R_1$. Let $\mathcal{H}_0 \subset \mathcal{H}$ denote the set hypotheses making at least $\frac{\epsilon m}{2}$ mistakes in $S''$. Now we divide $\mathcal{H}_0$ into two sub-classes as follows.

- Let $\mathcal{H}_1 = \{h \in \mathcal{H}_0 \mid h \text{ makes fewer than } \frac{\epsilon m}{4} \text{ mistakes in } R_2\}$. Let $\mathcal{F}_{S''}(\mathcal{H}_1) := \{f_{h,T} \mid h \in \mathcal{H}_1, T \subset S'', |T| = m\}$. For any $h$, if $S$ is correctly labeled, then $\text{err}_{\mathcal{G}S}(f_{h,S}) = 0$. Thus we have

$$\Pr(\exists h \in \mathcal{H}_1, \text{err}_S(h) = 0, \text{err}_{S'}(f_{h,S}) \geq \frac{\epsilon}{2})$$

$$\leq \Pr(\exists h \in \mathcal{H}_1, \text{err}_{\mathcal{G}S}(f_{h,S}) = 0, \text{err}_{S'}(f_{h,S}) \geq \frac{\epsilon}{2})$$

$$\leq \Pr(\exists f \in \mathcal{F}_{S''}(\mathcal{H}_1), \text{err}_{\mathcal{G}S}(f) = 0, \text{err}_{S'}(f) \geq \frac{\epsilon}{2}).$$

Since every $h \in \mathcal{H}_1$ makes fewer than $\frac{\epsilon m}{4}$ mistakes in $R_2$, for all $f \in \mathcal{F}_{S''}(\mathcal{H}_1)$, if $f$ makes at least $\frac{\epsilon m}{2}$ mistakes in $S''$, it must makes at least $\frac{\epsilon m}{4}$ mistakes in $R_1$. Similar to the case 1 in the proof of Theorem 1, for any $f \in \mathcal{F}_{S''}(\mathcal{H}_1)$, we let $X(f) \subset R_1$ denote a minimal set of examples in $R_1$ (breaking ties arbitrarily but in a fixed way) such that $f$ misclassify $X(f)$ and $|(\mathcal{G}X(f)) \cap S''_{\mathcal{X}}| \geq \frac{\epsilon m}{4}$ where $\mathcal{G}X(f) = \{gx \mid x \in X(f), g \in \mathcal{G}\}$ is the set of all examples lying in the orbits generated from $X(f)$. Let $K(f) = \mathcal{G}X(f)$ and $\mathcal{K} = \{K(f) \mid f \in \mathcal{F}_{S''}(\mathcal{H}_1)\}$. Notice that each example in $X(f)$ must belong to different orbits, otherwise it is not minimal. Besides, each orbit contains at least $\log^2 m$ examples from $S''_{\mathcal{X}}$. Hence, $|X(f)| \leq \frac{\epsilon m}{4 \log^2 m}$. Since there are at most $\frac{2m}{\log^2 m}$ orbits generated from $R_1$, we have $|\mathcal{K}| \leq \sum_{i=1}^{\frac{\epsilon m}{4 \log^2 m}} \binom{\frac{2m}{\log^2 m}}{i} \leq \left(\frac{8e}{\epsilon}\right)^{\frac{\epsilon m}{4 \log^2 m}}$. Since $f$ misclassify $X(f)$, all examples in $K(f)$ must go to $S'$ to guarantee $\text{err}_{\mathcal{G}S}(f) = 0$ and thus, we have

$$\Pr(\exists f \in \mathcal{F}_{S''}(\mathcal{H}_1), \text{err}_{\mathcal{G}S}(f) = 0, \text{err}_{S'}(f) \geq \frac{\epsilon}{2})$$

$$\leq \Pr(\exists f \in \mathcal{F}_{S''}(\mathcal{H}_1), K(f) \cap S_{\mathcal{X}} = \emptyset) = \Pr(\exists K \in \mathcal{K}, K \cap S_{\mathcal{X}} = \emptyset)$$

$$\leq \sum_{K \in \mathcal{K}} 2^{-\frac{\epsilon m}{4}} \leq \left(\frac{8e}{\epsilon}\right)^{\frac{\epsilon m}{4 \log^2 m}} \cdot 2^{-\frac{\epsilon m}{4}} = 2^{-\frac{\epsilon m}{4}(1 - \frac{\log(8e/\epsilon)}{\log^2 m})} \leq 2^{-\frac{\epsilon m}{8}},$$

when $m \geq \frac{8e}{\epsilon} + 4$.

- Let $\mathcal{H}_2 = \mathcal{H}_0 \setminus \mathcal{H}_1$. Now we will bound $\Pr(\exists h \in \mathcal{H}_2, \text{err}_S(h) = 0, \text{err}_{S'}(f_{h,S}) \geq \frac{\epsilon}{2})$. Similar to the case 2 in Theorem 1, every $h \in \mathcal{H}_2$ will make at least $\frac{\epsilon m}{4}$ mistakes in $R_2$. Since $\text{VC}_{\text{ao}}(\mathcal{H}, \mathcal{G}) = d$ and every orbit generated from $R_2$ contains fewer than $\log^2 m$ examples, the number of examples in $R_2$ that can be shattered by $\mathcal{H}$ is no greater than $d \log^2 m$. Thus, the number of ways labeling examples in $R_2$ is upper bounded by $(\frac{2em}{d})^{d \log^2 m}$ by Sauer's lemma. For any multi-subset $X \subset S''_{\mathcal{X}}$ and hypothesis $h$, we denote by $\widehat{M}_X(h)$ the number of instances in $X$ misclassified by $h$. Hence, we have

$$\Pr(\exists h \in \mathcal{H}_2, \text{err}_S(h) = 0, \text{err}_{S'}(f_{h,S}) \geq \frac{\epsilon}{2})$$

$$\leq \Pr(\exists h \in \mathcal{H}_2, \widehat{M}_{S_{\mathcal{X}} \cap R_2}(h) = 0, \widehat{M}_{S'_{\mathcal{X}} \cap R_2}(h) \geq \frac{\epsilon m}{4})$$

$$\leq (\frac{2em}{d})^{d \log^2 m} \cdot 2^{-\frac{\epsilon m}{4}} = 2^{-\frac{\epsilon m}{4} + d \log^2 m \log(2em/d)}$$

Combining the results for $\mathcal{H}_1$ and $\mathcal{H}_2$, we have

$$\Pr(B_{S,S'}) \leq 2^{-\frac{\epsilon m}{8}} + 2^{-\frac{\epsilon m}{4} + d \log^2 m \log(2em/d)} \leq \frac{\delta}{2},$$

when $m \geq \frac{8}{\epsilon}(d \log^2 m \log \frac{2em}{d} + \log \frac{4}{\delta} + e) + 4$. $\qquad \square$

*Proof of the upper bound of the 1-inclusion-graph predictor.* The algorithm is similar to the 1-inclusion-graph predictor in Theorem 4. For any $t \in \mathbb{N}$ and $S = \{(x_1, y_1), \ldots, (x_t, y_t)\}$, let $X_S$ be the set of different elements in $S_{\mathcal{X}}$ and $\mathcal{H}'(X_S) := \{h_{|X_S}| \forall x', x \in X_S, x' \in \mathcal{G}x \text{ implies } h(x') = h(x)\}$. Here $\mathcal{H}'(X_S)$ is different from $\mathcal{H}(X_S)$ defined in Theorem 4 in the sense that every hypothesis in $\mathcal{H}'(X_S)$ is not $(\mathcal{G}, X_S)$-invariant but only predict the observed examples in the same orbit in the same way. Note that $h^*_{|X_S}$ is in $\mathcal{H}'(X_S)$ if $S$ is realized by $h^*$. Let $Q_{\mathcal{H}'(X_S), X_S}$ be the function guaranteed by Eq (1) for the instance space $X_S$ and hypothesis class $\mathcal{H}'(X_S)$. Given a set $S$ of $t$ i.i.d. samples, we let $\mathcal{A}(S)$ be defined as $\mathcal{A}(S)$ be defined as $\mathcal{A}(S, x) = Q_{\mathcal{H}'(X_S \cup \{x\}), X_S \cup \{x\}}(S, x)$ if $\mathcal{H}'(X_S \cup \{x\})$ is nonempty and predicting arbitrarily if it is empty. Following the analogous proof of Theorem 4, we have

$$
\begin{aligned}
\mathbb{E}_{S \sim \mathcal{D}^t}\left[\mathrm{err}(\mathcal{A}(S))\right] =& \mathbb{E}_{(x_i, y_i)_{i \in [t+1]} \sim \mathcal{D}^{t+1}}\left[\mathbb{1}[\mathcal{A}(\{x_i, y_i\}_{i \in [t]}, x_{t+1}) \neq y_{t+1}]\right] \\
=& \frac{1}{(t+1)!} \sum_{\sigma \in \mathrm{Sym}(t+1)} \mathbb{E}\left[\mathbb{1}[\mathcal{A}(\{x_{\sigma(i)}, y_{\sigma(i)}\}_{i \in [t]}, x_{\sigma(t+1)}) \neq y_{\sigma(t+1)}]\right] \\
=& \mathbb{E}\left[\frac{1}{(t+1)!} \sum_{\sigma \in \mathrm{Sym}(t+1)} \mathbb{1}[\mathcal{A}(\{x_{\sigma(i)}, y_{\sigma(i)}\}_{i \in [t]}, x_{\sigma(t+1)}) \neq y_{\sigma(t+1)}]\right] \\
\leq& \frac{\mathbb{E}\left[\mathrm{VCdim}(\mathcal{H}'(\{x_i | i \in [t+1]\}))\right]}{t+1} \leq \frac{\mathrm{VC_{ao}}(\mathcal{H}, \mathcal{G})}{t+1}.
\end{aligned}
$$

Again, we use the same method as that in Theorem 4 to convert this algorithm, guaranteeing the expected error upper bounded by $\mathrm{VC_{ao}}(\mathcal{H}, \mathcal{G})$, into an algorithm with high probability $1 - \delta$. Specifically, the algorithm runs $\mathcal{A}$ for $\lceil \log(2/\delta) \rceil$ times, each time using a new sample of size $\lceil 4\mathrm{VC_{ao}}(\mathcal{H}, \mathcal{G})/\epsilon \rceil$. Then the algorithm selects the hypothesis from the outputs with the minimal error on a new sample of size $\lceil 32/\epsilon(\ln(2/\delta) + \ln(\lceil \log(2/\delta) \rceil + 1)) \rceil$. □

## H    Proof of Theorem 8

To prove the theorem, we will use a generalization bound for agnostic compression scheme by Graepel et al. (2005).

**Lemma 4** (Agnostic compression generalization bound). *There exists a finite numerical constant $c > 0$ such that, for any compression scheme $(\kappa, \rho)$, for any $n \in \mathbb{N}$ and $\delta \in (0, 1)$, for any distribution $\mathcal{D}$ on $\mathcal{X} \times \mathcal{Y}$, for $S \sim \mathcal{D}^n$, letting $B(S, \delta) := \frac{1}{n}(|\kappa(S)| \log(n) + \log(1/\delta))$, with probability at least $1 - \delta$, then*

$$
|\mathrm{err}_{\mathcal{D}}(\rho(\kappa(S))) - \mathrm{err}_S(\rho(\kappa(S)))| \leq c\sqrt{\mathrm{err}_S(\rho(\kappa(S)))B(S, \delta)} + cB(S, \delta).
$$

*Proof of the lower bound.* For the lower bound, our construction follows Ben-David and Urner (2014). For any $\mathcal{X}, \mathcal{H}$, let $A_1, \ldots, A_d$ be subsets of $\mathcal{X}$ such that the orbits of every two different elements $x, x' \in \cup_{i \in [d]} A_i$ are disjoint, i.e., $\forall x, x' \in \cup_{i \in [d]} A_i, \mathcal{G}x \cap \mathcal{G}x' = \emptyset$. We say $\mathcal{H}$ set-shatters $A_1, \ldots, A_d$ if for every binary vector $\mathbf{y} \in \mathcal{Y}^d$, there exists some $h_{\mathbf{y}} \in \mathcal{H}$ such that for all $i \in [d]$ and $x \in \mathcal{X}$, if $x \in A_i$ then $h_{\mathbf{y}}(x) = y_i$. Then by following Theorem 7 of Ben-David and Urner (2014), if $\mathcal{H}$ set-shatters $A_1, \ldots, A_d$ for some infinite subsets $A_1, \ldots, A_d$ of $\mathcal{X}$, the standard agnostic PAC sample complexity of learning $\mathcal{H}$ under deterministic labels for instance space being $\cup_{i \in [d]} A_i$ is lower bounded by $\frac{d}{\epsilon^2}$ for all $\delta < 1/32$. Since any data distribution $\mathcal{D}$ with $\mathcal{D}_{\mathcal{X}}(\cup_{i \in [d]} A_i) = 1$ is $\mathcal{G}$-invariant, the above lower bound also lower bounds $\mathcal{M}_{\mathrm{AG}}(\epsilon, \delta; \mathcal{H}, \mathcal{G})$ for all $\delta < 1/32$. Let $h(x) = 0$ for all $x \notin \cup_{i \in [k]} A_i$ and for all $h \in \mathcal{H}$, $\mathrm{VC_{ao}}(\mathcal{H}, \mathcal{G}) = d$. Thus, $\mathcal{M}_{\mathrm{AG}}(\epsilon, 1/64; \mathcal{H}, \mathcal{G}) = \Omega(\frac{\mathrm{VC_{ao}}(\mathcal{H}, \mathcal{G})}{\epsilon^2})$. The construction above works for any $d > 0$. □

*Proof of the upper bound of ERM-INV.* Let $d = \mathrm{VC_{ao}}(\mathcal{H}, \mathcal{G})$. Consider two sets $S$ and $S'$ of $m$ i.i.d. samples drawn from the data distribution $\mathcal{D}$ each. We denote $A_S$ the event of $\{\exists h \in \mathcal{H}, \mathrm{err}_{\mathcal{D}}(f_{h,S}) \geq \mathrm{err}_S(h) + \epsilon\}$ and $B_{S,S'} = \{\exists h \in \mathcal{H}, \mathrm{err}_{S'}(f_{h,S}) \geq \mathrm{err}_S(h) + \frac{\epsilon}{2}\}$. By Hoeffding bound, we have $\Pr(B_{S,S'}) \geq \Pr(A_S) \cdot \Pr(B_{S,S'}|A_S) \geq \frac{1}{2}\Pr(A_S)$ when $m \geq \frac{2}{\epsilon^2}$. The sampling process of $S$ and $S'$ is equivalent to drawing $2m$ i.i.d. samples and then randomly partitioning into $S$ and $S'$ of $m$ each. For any fixed $S''$, let us divide $S''$ into two categories in terms of the number of examples

in each orbit. Let $R_1 = \{x \mid |\mathcal{G}x \cap S''_{\mathcal{X}}| \geq \log^2 m\}$ and $R_2 = S''_{\mathcal{X}} \setminus R_1$. For any multi-subset $X \subset S''_{\mathcal{X}}$ and hypothesis $h$, we denote by $\widehat{M}_X(h)$ the number of instances in $X$ misclassified by $h$. Let $\mathcal{F}_{S''}(\mathcal{H}) := \{f_{h,T} \mid h \in \mathcal{H}, T \subset S'', |T| = m\}$. Then we have

$$\Pr(B_{S,S'})$$

$$= \Pr(\exists h \in \mathcal{H}, \mathrm{err}_{S'}(f_{h,S}) \geq \mathrm{err}_S(h) + \frac{\epsilon}{2})$$

$$\leq \Pr(\exists h \in \mathcal{H}, (\widehat{M}_{S'_{\mathcal{X}} \cap R_1}(f_{h,S}) \geq \widehat{M}_{S_{\mathcal{X}} \cap R_1}(h) + \frac{\epsilon m}{4}) \vee (\widehat{M}_{S'_{\mathcal{X}} \cap R_2}(f_{h,S}) \geq \widehat{M}_{S_{\mathcal{X}} \cap R_2}(h) + \frac{\epsilon m}{4}))$$

$$\leq \underbrace{\Pr(\exists h \in \mathcal{H}, \widehat{M}_{S'_{\mathcal{X}} \cap R_1}(f_{h,S}) \geq \widehat{M}_{S_{\mathcal{X}} \cap R_1}(h) + \frac{\epsilon m}{4})}_{(a)}$$

$$+ \underbrace{\Pr(\exists h \in \mathcal{H}, \widehat{M}_{S'_{\mathcal{X}} \cap R_2}(f_{h,S}) \geq \widehat{M}_{S_{\mathcal{X}} \cap R_2}(h) + \frac{\epsilon m}{4})}_{(b)}.$$

For the first term, since $\mathrm{err}_{\mathcal{G}S}(f_{h,S}) = 0$ according to the definition of $f_{h,S}$,

$$(a) \leq \Pr(\exists f \in \mathcal{F}_{S''}(\mathcal{H}), \widehat{M}_{S'_{\mathcal{X}} \cap R_1}(f) \geq \frac{\epsilon m}{4}, \mathrm{err}_{\mathcal{G}S}(f) = 0).$$

Again, similar to case 1 in Theorem 1 (also the first upper bound in Theorem 7), for any $f \in \mathcal{F}_{S''}(\mathcal{H})$, we let $X(f) \subset R_1$ denote a minimal set of examples in $R_1$ (breaking ties arbitrarily but in a fixed way) such that $f$ misclassify $X(f)$ and $|(\mathcal{G}X(f)) \cap S''_{\mathcal{X}}| \geq \frac{\epsilon m}{4}$ where $\mathcal{G}X(f) = \{gx \mid x \in X(f), g \in \mathcal{G}\}$ is the set of all examples lying in the orbits generated from $X(f)$. Let $K(f) = \mathcal{G}X(f)$ and $\mathcal{K} = \{K(f) \mid f \in \mathcal{F}_{S''}(\mathcal{H})\}$. Notice that each example in $X(f)$ must belong to different orbits, otherwise it is not minimal. Besides, each orbit contains at least $\log^2 m$ examples. Hence, $|X(f)| \leq \frac{\epsilon m}{4 \log^2 m}$. Since there are at most $\frac{2m}{\log^2 m}$ orbits generated from $R_1$, we have $|\mathcal{K}| \leq \sum_{i=1}^{\frac{\epsilon m}{4 \log^2 m}} \binom{\frac{2m}{\log^2 m}}{i} \leq \left(\frac{8e}{\epsilon}\right)^{\frac{\epsilon m}{4 \log^2 m}}$. Since $f$ misclassify $X(f)$, all examples in $K(f)$ must go to $S'$ to guarantee $\mathrm{err}_{\mathcal{G}S}(f) = 0$ and thus, we have

$$\Pr(\exists f \in \mathcal{F}_{S''}(\mathcal{H}), \widehat{M}_{S'_{\mathcal{X}} \cap R_1}(f) \geq \frac{\epsilon m}{4}, \mathrm{err}_{\mathcal{G}S}(f) = 0)$$

$$\leq \Pr(\exists f \in \mathcal{F}_{S''}(\mathcal{H}), K(f) \cap S_{\mathcal{X}} = \emptyset) = \Pr(\exists K \in \mathcal{K}, K \cap S_{\mathcal{X}} = \emptyset)$$

$$\leq \sum_{K \in \mathcal{K}} 2^{-\frac{\epsilon m}{4}} \leq \left(\frac{8e}{\epsilon}\right)^{\frac{\epsilon m}{4 \log^2 m}} \cdot 2^{-\frac{\epsilon m}{4}} = 2^{-\frac{\epsilon m}{4}(1 - \frac{\log(8e/\epsilon)}{\log^2 m})} \leq 2^{-\frac{\epsilon m}{8}},$$

when $m \geq \frac{8e}{\epsilon}$. For the second term, since $\widehat{M}_{S'_{\mathcal{X}} \cap R_2}(h) \geq \widehat{M}_{S'_{\mathcal{X}} \cap R_2}(f_{h,S})$, then we have

$$(b) \leq \Pr(\exists h \in \mathcal{H}, \widehat{M}_{S'_{\mathcal{X}} \cap R_2}(h) \geq \widehat{M}_{S_{\mathcal{X}} \cap R_2}(h) + \frac{\epsilon m}{4})$$

$$\leq \left(\frac{2em}{d}\right)^{d \log^2 m} \cdot e^{-2m'(\frac{\epsilon m}{8m'})^2} \tag{14}$$

$$\leq e^{-\frac{\epsilon^2 m}{32} + d \log^2 m \ln(2em/d)},$$

where Eq (14) adopts Hoeffding bound and Sauer's lemma (the number points in $R_2$ that can be shattered by $\mathcal{H}$ is at most $d \log^2 m$; otherwise $\mathrm{VC}_{\mathrm{ao}}(\mathcal{H}, \mathcal{G}) > d$) and $m' = \max_{h \in \mathcal{H}} M_{R_2}(h)$. By setting $m \geq \frac{32}{\epsilon^2}(d \log^2 m \ln(2em/d) + \ln(8/\delta) + 1) + \frac{8}{\epsilon} \log(8/\delta)$, we have $\Pr(A_S) \leq \delta/2$. By Hoeffding bound, with probability at least $1 - \delta/2$, $\mathrm{err}_S(h^*) \leq \mathrm{err}_{\mathcal{D}}(h^*) + \frac{\epsilon}{2}$. By a union bound, we have that with probability at least $1 - \delta$, $\mathrm{err}_{\mathcal{D}}(f_{h,S}) \leq \mathrm{err}_S(h) + \epsilon \leq \mathrm{err}_S(h^*) + \epsilon \leq \mathrm{err}_{\mathcal{D}}(h^*) + 3\epsilon/2$. $\qquad \square$

*Proof of the upper bound of the 1-inclusion-graph predictor.* Here we provide another algorithm based on the technique of reduction-to-realizable of David et al. (2016) and the 1-inclusion-graph predictor in the relaxed realizable setting. Following the argument in Theorem 7, given a sample $S = \{(x_1, y_1), \ldots, (x_n, y_n)\}$, let $X_S$ be the set of different elements in $S_{\mathcal{X}}$ and define

$\mathcal{H}'(X_S) := \{h_{|X_S}|\forall x', x \in X_S, x' \in \mathcal{G}x \text{ implies } h(x') = h(x)\}$. Then we define an algorithm $\mathcal{A}_0$ as

$$\mathcal{A}_0(S, x) = \begin{cases} Q_{\mathcal{H}'(X_S \cup \{x\}), X_S \cup \{x\}}(S, x) & \text{if } \mathcal{H}'(X_S \cup \{x\}) \neq \emptyset, \\ 0 & \text{o.w.} \end{cases} \tag{15}$$

where $Q_{\mathcal{H}'(X_S \cup \{x\}), X_S \cup \{x\}}$ is the function guaranteed by Eq (1) for hypothesis class $\mathcal{H}'(X_S \cup \{x\})$ and instance space $X_S \cup \{x\}$. In the relaxed realizable setting, if $S \sim \mathcal{D}^n$ and $x \sim \mathcal{D}_{\mathcal{X}}$, $\mathcal{H}'(X_S \cup \{x\}) \neq \emptyset$ a.s. as it contains $h^*_{|X_S \cup \{x\}}$. While in the agnostic setting, it is not the case. Even if $S \sim \mathcal{D}^n$ and $x \sim \mathcal{D}_{\mathcal{X}}$, $\mathcal{H}'(X_S \cup \{x\})$ could be empty as there might exist an instance $x' \in X_S \cap \mathcal{G}x$ such that no hypothesis in $\mathcal{H}$ labeling them in the same way.

Let $d = \mathrm{VC}_{\mathrm{ao}}(\mathcal{H}, \mathcal{G})$. If a sample $\{(x_1, y_1), \ldots, (x_{n+1}, y_{n+1})\}$ is realizable by $\mathcal{H}$, then by Lemma 1, we have

$$\frac{1}{(n+1)!} \sum_{\sigma \in \mathrm{Sym}(n+1)} \mathbb{1}[\mathcal{A}_0(\{x_{\sigma(i)}, y_{\sigma(i)}\}_{i \in [n]}, x_{\sigma(n+1)}) \neq y_{\sigma(n+1)}] \leq \frac{d}{n+1}. \tag{16}$$

Now we use $\mathcal{A}_0$ as a weak learner to construct a compression scheme by following the construction David et al. (2016). Given a training set $S_{\mathrm{trn}} = \{(x_1, y_1), \ldots, (x_m, y_m)\}$, let $R$ denote the largest submulitset of $S_{\mathrm{trn}}$ that is realizable w.r.t. $\mathcal{H}$. If $|R| = 0$, then define $\widehat{h}$ as the all-0 function $\widehat{h}(x) = 0$. Otherwise, if $|R| > 0$, for any distribution $\mathcal{P}$ on $R$, by Eq (16), we have that

$$\mathbb{E}_{S \sim \mathcal{P}^{3d}}\left[\mathrm{err}_{\mathcal{P}}(\mathcal{A}_0(S))\right]$$

$$= \mathbb{E}_{(x_i, y_i)_{i \in [3d+1]} \sim \mathcal{P}^{3d}}\left[\frac{1}{(3d+1)!} \sum_{\sigma \in \mathrm{Sym}(3d+1)} \mathbb{1}[\mathcal{A}_0(\{x_{\sigma(i)}, y_{\sigma(i)}\}_{i \in [3d]}, x_{\sigma(3d+1)}) \neq y_{\sigma(3d+1)}]\right]$$

$$\leq \frac{1}{3}.$$

Hence, there exists $S_{\mathcal{P}} \in R^{3d}$ with $\mathrm{err}_{\mathcal{P}}(\mathcal{A}_0(S_{\mathcal{P}})) \leq 1/3$. Thus, the algorithm $\mathcal{A}_0$ can serve as a weak learner. By Lemma 2, for $T = \lceil c \log |R| \rceil$ (for a numerical constant $c > 0$), there exist $S_1, \ldots, S_T \in R^{3d}$ such that, letting $\widehat{h}(\cdot) = \mathrm{Majority}(\mathcal{A}_0(S_1)(\cdot), \ldots, \mathcal{A}_0(S_T)(\cdot))$, we have $\mathrm{err}_R(\widehat{h}) = 0$. Thus, $\mathrm{err}_{S_{\mathrm{trn}}}(\widehat{h}) \leq \inf_{h \in \mathcal{H}} \mathrm{err}_{S_{\mathrm{trn}}}(h)$. Here $\widehat{h}$ is the output of the compression scheme that selects $\kappa(S_{\mathrm{trn}}) = (S_1, \ldots, S_T)$ and $\rho(\kappa(S_{\mathrm{trn}})) = \widehat{h}$. By Lemma 4, with probability at least $1 - \delta/2$,

$$\mathrm{err}_{\mathcal{D}}(\widehat{h}) \leq \mathrm{err}_{S_{\mathrm{trn}}}(\widehat{h}) + c'\sqrt{\mathrm{err}_{S_{\mathrm{trn}}}(\widehat{h})B(S_{\mathrm{trn}}, \delta/2)} + c'B(S_{\mathrm{trn}}, \delta/2),$$

for a numerical constant $c' > 0$. Following the same argument of David et al. (2016) (Lemma 3.2), we have that

$$\Pr_{S_{\mathrm{trn}} \sim \mathcal{D}^m}\left(\mathrm{err}_S(\widehat{h}) \geq \inf_{h \in \mathcal{H}} \mathrm{err}_{\mathcal{D}}(h) + \sqrt{\log(2/\delta)/m}\right) \leq \delta/2.$$

By taking a union bound, then we have that with probability at least $1 - \delta$,

$$\mathrm{err}_{\mathcal{D}}(\widehat{h}) \leq \inf_{h \in \mathcal{H}} \mathrm{err}_{\mathcal{D}}(h) + c'\sqrt{\left(\inf_{h \in \mathcal{H}} \mathrm{err}_{\mathcal{D}}(h) + \sqrt{\log(2/\delta)/m}\right)\frac{1}{m}(3d\lceil c \log|R|\rceil \log m + \log 1/\delta)}$$

$$+ \frac{c'}{m}(3d\lceil c \log|R|\rceil \log m + \log(1/\delta)).$$

Hence,

$$\mathcal{M}_{\mathrm{AG}}(\epsilon, \delta; \mathcal{H}, \mathcal{G}) = O\left(\frac{d}{\epsilon^2}\log^2\left(\frac{d}{\epsilon}\right) + \frac{1}{\epsilon^2}\log(1/\delta)\right).$$

$\square$

# I  Adaptive algorithms

For any hypothesis $h \in \mathcal{H}$, we say $h$ is $(1 - \eta)$-invariant over the distribution $\mathcal{D}_{\mathcal{X}}$ for some $\eta \in [0, 1]$ if $\mathbb{P}_{x \sim \mathcal{D}_{\mathcal{X}}} (\exists x' \in \mathcal{G}x, h(x') \neq h(x)) = \eta$. We call $\eta(h) = \eta$ the invariance parameter of $h$ with respect to $\mathcal{D}_{\mathcal{X}}$. In the relaxed realizable setting, the problem degenerates into the invariantly realizable setting when $\eta(h^*) = 0$. This implies that we can benefit from transformation invariances more when $\eta(h^*)$ is smaller. The case is similar in the agnostic setting. In this section, we discuss adaptive learning algorithms for different levels of invariance of the target function.

## I.1  An adaptive algorithm in the relaxed realizable setting

There might exist more than one hypotheses in $\mathcal{H}$ with zero error and we let the target function $h^*$ be the one with the smallest invariance parameter (breaking ties arbitrarily). For any multiset $X$ in $\mathcal{X}$, for any hypothesis $h$, denote $h_{|X}$ the restriction of $h$ on the set of different elements in $X$. Then we introduce a distribution-dependent dimension as follows.

**Definition 8** (approximate $(1 - \eta)$-invariant VC dimension). *For any $\eta \in [0, 1]$ and finite multisubset $X \subset \mathcal{X}$, let $\mathcal{H}^{\eta}(X) := \{h_{|X} | \frac{1}{|X|} \sum_{x \in X} \mathbb{1}[\exists x' \in \mathcal{G}x, h(x') \neq h(x)] \leq \eta \wedge \forall x, x' \in \mathcal{G}x \cap X, h(x') = h(x)\}$. For any $m \in \mathbb{N}$, marginal distribution $\mathcal{D}_{\mathcal{X}}$ and target function $h^*$, the approximate $(1 - \eta)$-invariant VC dimension is defined as*

$$\mathrm{VC_o}^{\eta}(m, h^*, \mathcal{H}, \mathcal{G}, \mathcal{D}_{\mathcal{X}}) := \mathbb{E}_{X \sim \mathcal{D}_{\mathcal{X}}^{m+1}} \left[ \mathrm{VCdim}(\mathcal{H}^{\eta}(X)) | h^*_{|X} \in \mathcal{H}^{\eta}(X) \right],$$

*when $\Pr(h^*_{|X} \in \mathcal{H}^{\eta}(X)) > 0$ and $\mathrm{VC_o}^{\eta}(m, h^*, \mathcal{H}, \mathcal{G}, \mathcal{D}_{\mathcal{X}}) = 0$ when $\Pr(h^*_{|X} \in \mathcal{H}^{\eta}(X)) = 0$. By taking supremum over $m$,*

$$\mathrm{VC_o}^{\eta}(h^*, \mathcal{H}, \mathcal{G}, \mathcal{D}_{\mathcal{X}}) = \sup_{m \in \mathbb{N}} \mathrm{VC_o}^{\eta}(m, h^*, \mathcal{H}, \mathcal{G}, \mathcal{D}_{\mathcal{X}}).$$

In this definition, all hypotheses in $\mathcal{H}^{\eta}(X)$ need to satisfy two constraints: a) the empirical invariance parameter is less than or equal to $\eta$, and b) the prediction over instances in $X$ is invariant over orbits. Here the second constraint arises due to the fact that $h^*$ satisfies this constraint. Note that $\mathrm{VC_o}^{\eta}(h^*, \mathcal{H}, \mathcal{G}, \mathcal{D}_{\mathcal{X}})$ is monotonic increasing in $\eta$. For all $\mathcal{D}_{\mathcal{X}}, h^*$, we have $\mathrm{VC_o}^0(h^*, \mathcal{H}, \mathcal{G}, \mathcal{D}_{\mathcal{X}}) \leq \mathrm{VC_o}(\mathcal{H}, \mathcal{G})$ and $\mathrm{VC_o}^{\eta}(h^*, \mathcal{H}, \mathcal{G}, \mathcal{D}_{\mathcal{X}}) \leq \mathrm{VC_{ao}}(\mathcal{H}, \mathcal{G})$ for all $\eta \in [0, 1]$. We will use the approximate $(1 - \eta)$-invariant VC dimension to characterize the sample complexity dependent on $\eta$. Ideally, it is heuristic to adopt a notion like $\mathrm{VC_{ao}}(\{h \in \mathcal{H} | \eta(h) = \eta(h^*)\})$. But this is impossible to achieve as we cannot obtain an accurate estimate of $\eta(h^*)$ via finite data points.

**Proposition 1.** *If it is known that $\eta(h^*) \leq \eta$ for some $\eta \in [0, 1]$, for any training sample size $m \in \mathbb{N}$, for any $\Delta \geq \sqrt{\frac{\ln(n+1)}{2(n+1)}}$ with $n = \Theta(\frac{m}{\log(1/\delta)})$, there is an algorithm achieving error $O(\frac{\mathrm{VC_o}^{\eta+\Delta}(h^*, \mathcal{H}, \mathcal{G}, \mathcal{D}_{\mathcal{X}}) \log(1/\delta)}{m})$ with probability at least $1 - \delta$.*

Given a sample $S = \{(x_1, y_1), \ldots, (x_n, y_n)\}$ and a test instance $x$, let $X_S$ denote the set of different elements in $S_{\mathcal{X}}$ and $X = \{x_1, \ldots, x_n, x\}$ the multiset of all unlabeled instances from both the training set and the test instance. The algorithm $\mathcal{A}$ is defined by

$$\mathcal{A}(S, x) = Q_{\mathcal{H}^{\eta+\Delta}(X), X_S \cup \{x\}}(S, x),$$

where $Q_{\mathcal{H}^{\eta+\Delta}(X), X_S \cup \{x\}}$ is the function guaranteeed by Eq (1) for the instance space $X_S \cup \{x\}$ and the concept class $\mathcal{H}^{\eta+\Delta}(X)$. Similar to Theorem 4, algorithm $\mathcal{A}$ can achieve expected error $\frac{\mathrm{VC_o}^{\eta+\Delta}(h^*, \mathcal{H}, \mathcal{G}, \mathcal{D}_{\mathcal{X}}) + 1}{n+1}$. Then by following the same confidence boosting argument, we run $\mathcal{A}$ for $\lceil \log(2/\delta) \rceil$ times on independent new samples and select the output hypothesis with minimum error a new sample. The details of the algorithm and the proof of Proposition 1 are deferred to Appendix J.

In the more general case where $\eta$ is unknown, we build an algorithm based on the algorithm for known $\eta$ above. Denote $\mathcal{A}_{\eta, \Delta}$ the algorithm satisfying the guarantee in Proposition 1 with probability $1 - \delta/2$ for hyperparameters $\eta$ and $\Delta$. Then we divide $[0, 1]$ into uniform intervals and then search for the interval where $\eta(h^*)$ lies in. The detailed algorithm is provided in Algorithm 1.

**Theorem 9.** *Set $\Delta = \sqrt{\frac{\ln(n+1)}{2(n+1)}}$ for $n = \Theta(\frac{m}{\log(m) \log(1/\delta)})$ and $|S_1| = \Theta(\frac{m}{\log m})$. Let $i^* \geq 0$ be the smallest integer $i$ such that $\max((2i - 1)\Delta, 0) \geq \eta(h^*)$. Then Algorithm 1 achieves error $O(\frac{\mathrm{VC_o}^{2i^*\Delta}(h^*, \mathcal{H}, \mathcal{G}, \mathcal{D}_{\mathcal{X}}) \log(1/\delta) \log(m)}{m})$ with probability at least $1 - \delta$.*

---

**Algorithm 1** An adaptive algorithm in the relaxed realizable setting

---
1: Input: a labeled sample $S_{\mathrm{trn}}$ of size $m$, $m_1 \in \mathbb{N}$, $\Delta \in [0,1]$
2: Randomly partition $S$ into $S_1$ with $|S_1| = m_1$ and $S_2$ with $|S_2| = m - m_1$
3: **for** $i = 0, 1, \ldots, \lceil 1/(2\Delta) \rceil$ **do**
4:    Let $h_i = \mathcal{A}_{(2i-1)\Delta, \Delta}(S_1)$
5:    Return $\widehat{h} = \arg\min_{h \in \{h_i | i \in \{0, \ldots, \lceil 1/(2\Delta) \rceil\}\}} \mathrm{err}_{S_2}(h)$
6: **end for**

---

---

**Algorithm 2** An adaptive algorithm in the agnostic case

---
1: Input: a unlabeled data set $U$ of size $u$ and a labeled data set $S$ of size $m$, $m_1 \in \mathbb{N}$, $\Delta > 0$
2: Randomly divide $S$ into $S_1$ of size $m_1$ and $S_2$ of size $m - m_1$
3: Use $U$ to partition $\mathcal{H}$ into $\widehat{\mathcal{H}}_1, \ldots, \widehat{\mathcal{H}}_K$ with $\widehat{\mathcal{H}}_i = \{h \in \mathcal{H} | \frac{1}{u} \sum_{x \in U} \mathbb{1}[\exists x' \in \mathcal{G}x, h(x') \neq h(x)] \in (2(i-1)\Delta, 2i\Delta]\}$ and $K = \lceil \frac{1}{2\Delta} \rceil$
4: **for** $i = 0, 1, 2, \ldots, K$ **do**
5:    Run the algorithm $\mathcal{A}$ in Theorem 8 over $S_1$ for hypothesis class $\widehat{\mathcal{H}}_i$ and output $h_i$
6:    Return $\widehat{h} = \arg\min_{h \in \{h_i | i \in \{0, \ldots, K\}\}} \mathrm{err}_{S_2}(h)$
7: **end for**

---

The proof is deferred to Appendix K. The algorithm above perform close to optimally in both the invariant realizable setting and the relaxed realizable setting. Intuitively, the algorithm above outperforms PAC-optimal algorithms in Theorem 7 when $\eta(h^*)$ is small. Below is an example showing the advantage of this adaptive algorithm in the extreme case of $\eta(h^*) = 0$.

**Example 6.** *Consider the construction of $\mathcal{H}_d, \mathcal{G}_d, h^*, \mathcal{D}$ in the proof of Theorem 2. In this example, we have $i^* = 0$ and $\mathcal{H}^0(X) = \{\mathbf{1}\}$ for any $X$. Thus, $\mathrm{VC_o}^0(h^*, \mathcal{H}, \mathcal{G}, \mathcal{D}_\mathcal{X}) = 0$. Hence, the algorithm above only requires sample complexity $O(1)$ to output a zero-error predictor. However, algorithms only caring the worst case upper bound may require much more samples. For example, ERM-INV predicts exactly the same as standard ERM in this example and thus, it requires $\Omega(\frac{\mathrm{VC_{ao}}(\mathcal{H}, \mathcal{G})}{\epsilon})$ samples to achieve $\epsilon$ error.*

**Open question:** It is unclear whether this adaptive algorithm is optimal and whether the approximate $(1-\eta)$-invariant VC dimension is the best $\eta$-dependent measure to characterize the sample complexity.

### I.2 An adaptive algorithm in the agnostic setting

In the agnostic setting, we assume that there exists an optimal hypothesis $h^* \in \mathcal{H}$ such that $\mathrm{err}(h^*) = \inf_{h \in \mathcal{H}} \mathrm{err}(h)$. Then similar to the realizable setting, we can design algorithms that adapt to $\eta(h^*)$. However, it is more challenging to design an adaptive algorithm in the agnostic setting than in the relaxed realizable setting. One of the most direct ideas is to combine agnostic compression scheme with the adaptive algorithm in the realizable setting. One possible way of combination is finding the largest realizable subset of the data and applying the adaptive algorithm in the relaxed realizable setting. However, this does not work since the realizable subset is not i.i.d. and the empirical invariance parameter calculated based on this subset is biased. Another possible way of combination is calculating the empirical invariance parameter over the whole data set, reducing the hypothesis class based on this empirical value and then run the compression scheme in Theorem 8 based on this reduced hypothesis class. This does not work either because the predictor depends on the whole data set now and the compression size is too large. Hence, there is a significant barrier that has arisen as a result of estimating the invariance parameter while obtaining low error at the same time.

To get around this obstacle, we provide an approach of using two independent data sets. Specifically, we partition the hypothesis class into subclasses with different empirical invariance parameters based on a data set first. Notice that in this step, we only need an unlabeled data set. Then we run the compression scheme in Theorem 8 for each subclass and return the one with the small validation error. The detailed algorithm is presented in Algorithm 2.

**Theorem 10.** *For each $h \in \mathcal{H}$, the invariance indicator function of $h$ is a mapping $\iota_h : \mathcal{X} \mapsto \{0,1\}$ such that $\iota_h(x) = \mathbb{1}[\exists x' \in \mathcal{G}x, h(x') \neq h(x)]$. Denote $\mathcal{I} = \{\iota_h | h \in \mathcal{H}\}$ the set of invariance indicator functions for all $h \in \mathcal{H}$. Then for any $\Delta \in (0,1]$, Algorithm 2 can achieve*

$$\mathrm{err}(\widehat{h}) \leq \mathrm{err}(h^*) + O\left(\sqrt{\frac{\mathrm{VC}_{\mathrm{ao}}(\mathcal{H}^*)\log^2 m + \log(1/\delta) + \log(1/\Delta)}{m}}\right), \textit{ where } \mathcal{H}^* = \{h|\eta(h) \in (\eta(h^*) -$$

$$2\Delta - 2\Delta', \eta(h^*) + 2\Delta + 2\Delta')\} \textit{ with } \Delta' = \Theta(\sqrt{\frac{\mathrm{VCdim}(\mathcal{I})\log(u) + \log(1/\delta)}{u}}).$$

The detailed proof of Theorem 10 is deferred to Appendix L. The upper bound in Theorem 10 depends on $\mathrm{VCdim}(\mathcal{I})$, which can be arbitrarily larger than $\mathrm{VCdim}(\mathcal{H})$. For example, for any $d > 0$, let $\mathcal{X} = \{0,1\}^d \times [d]$. For each $\mathbf{b} \in \{0,1\}^d$, define a hypothesis $h_{\mathbf{b}}$ by letting $h_{\mathbf{b}}((\mathbf{b}, i)) = b_i$ and $h_{\mathbf{b}}(x) = 0$ for all other $x \in \mathcal{X}$. Let the hypothesis class $\mathcal{H} = \{h_{\mathbf{b}}|\mathbf{b} \in \{0,1\}^d\}$. Then it is direct to check that $\mathrm{VCdim}(\mathcal{H}) = 1$ but $\mathrm{VCdim}(\mathcal{I}) = d$. It is unclear how to design an adaptive algorithm with theoretical guarantee independent of $\mathrm{VCdim}(\mathcal{I})$. The $\eta$-dependent dimension we adopt in the agnostic setting is different from that in the relaxed realizable setting. It is also unclear what is the best way to characterize the dependence on $\eta$ in the agnostic setting.

## J  Proof of Proposition 1

*Proof.* The proof follows that of Theorem 4. Given a training sample of size $m$, let $n$ be the largest integer such that $m \geq n\lceil\log(2/\delta)\rceil + \lceil 8(n+1)(\ln(2/\delta) + \ln(\lceil\log(2/\delta)\rceil + 1))\rceil$. For any sample $S = \{(x_1, y_1), \ldots, (x_n, y_n)\}$ and a test instance $x_{n+1} = x$, if $h^*$ is in $\mathcal{H}^{\eta+\Delta}(X)$, by Lemma 1, we have that

$$\frac{1}{(n+1)!}\sum_{\sigma \in \mathrm{Sym}(n+1)} \mathbb{1}[\mathcal{A}(\{x_{\sigma(i)}, y_{\sigma(i)}\}_{i\in[n]}, x_{\sigma(n+1)}) \neq y_{\sigma(n+1)}] \leq \frac{\mathrm{VCdim}(\mathcal{H}^{\eta+\Delta}(X))}{n+1}.$$

Thus,

$$\mathbb{E}_{S\sim\mathcal{D}^n}\left[\mathrm{err}(\mathcal{A}(S))\right]$$

$$=\mathbb{E}_{(x_i,y_i)_{i\in[n+1]}\sim\mathcal{D}^{n+1}}\left[\mathbb{1}[\mathcal{A}(\{x_i, y_i\}_{i\in[n]}, x_{n+1}) \neq y_{n+1}]\right]$$

$$=\frac{1}{(n+1)!}\sum_{\sigma\in\mathrm{Sym}(n+1)}\mathbb{E}\left[\mathbb{1}[\mathcal{A}(\{x_{\sigma(i)}, y_{\sigma(i)}\}_{i\in[n]}, x_{\sigma(n+1)}) \neq y_{\sigma(n+1)}]\right]$$

$$=\mathbb{E}\left[\frac{1}{(n+1)!}\sum_{\sigma\in\mathrm{Sym}(n+1)}\mathbb{1}[\mathcal{A}(\{x_{\sigma(i)}, y_{\sigma(i)}\}_{i\in[n]}, x_{\sigma(n+1)}) \neq y_{\sigma(n+1)}]\right]$$

$$\leq\mathbb{E}\left[\frac{1}{(n+1)!}\sum_{\sigma\in\mathrm{Sym}(n+1)}\mathbb{1}[\mathcal{A}(\{x_{\sigma(i)}, y_{\sigma(i)}\}_{i\in[n]}, x_{\sigma(n+1)}) \neq y_{\sigma(n+1)}]\,\Big|\,h^*_{|X} \in \mathcal{H}^{\eta+\Delta}(X)\right]$$

$$\quad\cdot\Pr(h^*_{|X} \in \mathcal{H}^{\eta+\Delta}(X)) + \Pr(h^*_{|X} \notin \mathcal{H}^{\eta+\Delta}(X))$$

$$\leq\frac{\mathbb{E}_{X\sim\mathcal{D}_{\mathcal{X}}^{n+1}}\left[\mathrm{VCdim}(\mathcal{H}^{\eta+\Delta}(X))|h^*_{|X} \in \mathcal{H}^{\eta+\Delta}(X)\right]}{n+1}\Pr(h^*_{|X} \in \mathcal{H}^{\eta+\Delta}(X))$$

$$\quad+ \Pr(h^*_{|X} \notin \mathcal{H}^{\eta+\Delta}(X))$$

$$\leq\frac{\mathrm{VC_o}^{\eta+\Delta}(h^*, \mathcal{H}, \mathcal{G}, \mathcal{D}_{\mathcal{X}})}{n+1}\Pr(h^*_{|X} \in \mathcal{H}^{\eta+\Delta}(X)) + \Pr(h^*_{|X} \notin \mathcal{H}^{\eta+\Delta}(X))$$

$$\leq\frac{\mathrm{VC_o}^{\eta+\Delta}(h^*, \mathcal{H}, \mathcal{G}, \mathcal{D}_{\mathcal{X}}) + 1}{n+1}, \tag{17}$$

where Eq (17) holds due to $\Pr(h^*_{|X} \notin \mathcal{H}^{\eta+\Delta}(X)) \leq \Pr(\frac{1}{|X|}\sum_{x\in X}\mathbb{1}[\exists x' \in \mathcal{G}x, h^*(x') \neq h^*(x)] - \eta(h^*) \geq \Delta) \leq \exp(-2(n+1)\Delta^2)$ by Hoeffding bound.

Then we again follow the classic technique to boost the confidence. The algorithm runs $\mathcal{A}$ for $\lceil\log(2/\delta)\rceil$ times, each time using a new sample $S_i$ of size $n$ for $i = 1, \ldots, \lceil\log(2/\delta)\rceil$. Let $h_i = \mathcal{A}(S_i)$ and then selects the hypothesis $\widehat{h}$ from $\{h_i|i \in [\lceil\log(2/\delta)\rceil]\}$ with the minimal error on a new sample $S_0$ of size $t = \lceil 8(n+1)(\ln(2/\delta) + \ln(\lceil\log(2/\delta)\rceil + 1))\rceil$.

Denote $\epsilon = \frac{\mathrm{VC_o}^{\eta+\Delta}(h^*, \mathcal{H}, \mathcal{G}, \mathcal{D}_{\mathcal{X}}) + 1}{n+1}$. For each $i$, by Eq (17), we have $\mathbb{E}\left[\mathrm{err}(h_i)\right] \leq \epsilon$. By Markov's inequality, with probability at least $\frac{1}{2}$, $\mathrm{err}(h_i) \leq 2\epsilon$. Since $h_i$ are independent, we have that with

probability at least $1 - \frac{\delta}{2}$, at least one of $\{h_i | i \in [\lceil \log(2/\delta) \rceil]\}$ has error smaller than $2\epsilon$. Then by Chernoff bound, for each $i$, on the event $\text{err}(h_i) \leq 2\epsilon$,

$$\Pr(\text{err}_{S_0}(h_i) > 3\epsilon | h_i) < e^{-\frac{t\epsilon}{6}}.$$

Also, on the event $\text{err}(h_i) > 4\epsilon$,

$$\Pr(\text{err}_{S_0}(h_i) \leq 3\epsilon | h_i) \leq e^{-\frac{t\epsilon}{8}}.$$

Thus, by the law of total probability and a union bound, with probability at least $1 - \frac{\delta}{2}$, if any $i$ has $\text{err}(h_i) \leq 2\epsilon$, then the returned the hypothesis $\widehat{h}$ has $\text{err}(\widehat{h}) \leq 4\epsilon$. By a union bound, the proof is completed. $\square$

## K  Proof of Theorem 9

*Proof.* Given a training sample of size $m$, let $|S_1| = m_1$ and $|S_2| = m_2 = m - m_1$. The values of $m_1, m_2$ are determined later. Let $n$ be the largest integer such that $m_1 \geq n \lceil \log(2/\delta) \rceil + \lceil 8(n+1)(\ln(2/\delta) + \ln(\lceil \log(2/\delta) \rceil + 1)) \rceil$. Let $\Delta = \sqrt{\frac{\ln(n+1)}{2(n+1)}}$ and then the number of rounds we run $\mathcal{A}_{\eta,\Delta}$ as a subroutine is upper bounded by $\lceil 1/(2\Delta) \rceil + 1 = \left\lceil \sqrt{(n+1)/(2\ln(n+1))} \right\rceil + 1 \leq m$. According to Proposition 1, there is a numerical constant $c > 0$ such that with probability $1 - \delta/2$, $\text{err}(h_{i^*}) \leq \frac{c\text{VC}_{\text{o}}{}^{2i^*\Delta}(h^*, \mathcal{H}, \mathcal{G}, \mathcal{D}_{\mathcal{X}}) \ln(2/\delta)}{m_1}$. Denote $\epsilon = \frac{c\text{VC}_{\text{o}}{}^{2i^*\Delta}(h^*, \mathcal{H}, \mathcal{G}, \mathcal{D}_{\mathcal{X}}) \ln(2/\delta)}{m_1}$. By Chernoff bound, for each $i \neq i^*$, on the event $\text{err}(h_i) > 4\epsilon$,

$$\Pr(\text{err}_{S_2}(h_i) \leq 2\epsilon | h_i) \leq e^{-\frac{m_2\epsilon}{2}}.$$

Also, on the event $\text{err}(h_{i^*}) \leq \epsilon$,

$$\Pr(\text{err}_{S_2}(h_{i^*}) > 2\epsilon | h_{i^*}) < e^{-\frac{m_2\epsilon}{3}}.$$

Let $m_1 = \frac{cm}{3\ln m + c}$ and $m_2 = \frac{3m \ln m}{3 \ln m + c}$. Then with probability at least $1 - me^{-m_2\epsilon/3} \geq 1 - \delta/2$, if $\text{err}(h_{i^*}) \leq \epsilon$, then the returned classifier has error smaller than $4\epsilon$. By taking a union bound, the proof is completed. $\square$

## L  Proof of Theorem 10

*Proof.* Given a labeled data set $S$ of size $m$, let $|S_1| = m_1$ and $|S_2| = m_2 = m - m_1$. The values of $m_1, m_2$ are determined later. Let $\widehat{i}$ be the $i \in [K]$ such that $h^* \in \widehat{\mathcal{H}}_i$. By Theorem 8, we have that there exists a numerical constant $c > 0$ such that with probability at least $1 - \delta/3$,

$$\text{err}(h_{\widehat{i}}) \leq \text{err}(h^*) + c\sqrt{\frac{\text{VC}_{\text{ao}}(\widehat{\mathcal{H}}_{\widehat{i}}) \log^2 m_1 + \log(1/\delta)}{m_1}}.$$

Then by Hoeffding bound, for any $\epsilon > 0$, for each $i \in K$, with probability at least $1 - \delta/(3K)$,

$$|\text{err}_{S_2}(h_i) - \text{err}(h_i)| \leq \epsilon,$$

when $m_2 \geq \frac{\ln(6K/\delta)}{2\epsilon^2}$. Then by taking a union bound, with probability at least $1 - \delta/3$, for all $i \in K$ we have

$$|\text{err}_{S_2}(h_i) - \text{err}(h_i)| \leq \epsilon.$$

Hence, with probability at least $1 - 2\delta/3$,

$$\text{err}(\widehat{h}) \leq \text{err}(h_{\widehat{i}}) + 2\epsilon \leq \text{err}(h^*) + c\sqrt{\frac{\text{VC}_{\text{ao}}(\widehat{\mathcal{H}}_{\widehat{i}}) \log^2 m_1 + \log(1/\delta)}{m_1}} + 2\epsilon.$$

By uniform convergence bound, with probability at least $1 - \delta/3$,

$$\left| \frac{1}{u} \sum_{x \in U} \mathbb{1}[\exists x' \in \mathcal{G}x, h(x') \neq h(x)] - \eta(h) \right| \leq \Delta', \forall h \in \mathcal{H}.$$

It follows that $\widehat{\mathcal{H}}_{\widehat{i}} \subset \mathcal{H}^*$ and thus, $\text{VC}_{\text{ao}}(\widehat{\mathcal{H}}_{\widehat{i}}) \leq \text{VC}_{\text{ao}}(\mathcal{H}^*)$. The proof is completed by letting $\epsilon = \sqrt{\frac{\text{VC}_{\text{ao}}(\mathcal{H}^*) + \log(1/\delta)}{m_1}}$ and $m_1 = \frac{2(\text{VC}_{\text{ao}}(\mathcal{H}^*) + \log(1/\delta))m}{2(\text{VC}_{\text{ao}}(\mathcal{H}^*) + \log(1/\delta)) + \ln(6K/\delta)}$. $\square$