# OpenReview forum: "A Theory of PAC Learnability under Transformation Invariances"
_NeurIPS.cc/2022/Conference — NeurIPS 2022 Accept_

### Official Review · Reviewer_vPzm · 2022-06-27

**Rating:** 7
**Confidence:** 3
**Soundness:** 3 good
**Presentation:** 2 fair
**Contribution:** 2 fair

**Summary:**

Modern computer vision models are trained to be robust to various kinds of transformations such as rotation, tilt, reflection of input images. This is usually achieved with data augmentation when the transformed images are explicitly added to the training set. In an attempt to have a theory that takes into account this feature of modern approaches to learning the authors propose to expand the classical PAC framework by introducing transformation invariant hypotheses and distributions. Realizable and agnostic cases are considered. The modifications of the VC dimension are given, which characterize learnability in different realizability modes. Optimal (in sample complexity) learning algorithms are discussed.

**Questions:**

Perhaps I missed it in the papaer, but I have this question: if we perform data augmentation under standard PAC learning, but define the sample size as the number of examples in the original (non-augmented) dataset, how does this compare with your estimate of the sample complexity $\tilde{O}\left(\frac{\text{VC}_{\text{ao}}(\mathcal{H},\mathcal{G})}{\epsilon}\right)$ in the invariantly realizable case?

**Limitations:**

It seems to me that the proposed theory has a limited scope. See item 1 in Strengths and Weaknesses above.

**Strengths And Weaknesses:**

1) As far as I know, data augmentation techniques are rather simple in the case of image data, but not so simple in the case of textual data. Say, to implement the technique that the authors mentioned at the beginning of the paper (replacement of noun phrases), some external resources will be required (for example, an already trained constituent parser) which we may not assume to be readily available for a wide variety of languages. Therefore, it seems to me that the proposed theory has a limited scope.

2) The proposed theory does not provide a simple characterization of realizable and agnostic learnability, as does the VC dimension in classical PAC learning. In the proposed theory, different modes of realizability entail different characterizations of learnability.

3) The proposed theory does not provide such a simple principle as the ERM in classical PAC learning. The optimal learning algorithm is again conditioned by the realizability mode. Moreover, in some cases the optimal algorithm is transductive (based on a 1-inclusion-graph predictor), i.e. to calculate the label on any new test point, the entire training set will be used (as it is, for example, in the k nearest neighbors algorithm), which calls into question the computational efficiency of learning.

---

> ### Author Response · Authors · 2022-08-02
> **Response to Reviewer vPzm**
>
> Thanks for your review!
>
> To your question on application in textual data: To clarify, this work considers the setting where the transformations and how to apply these transformations are known. The difficulty of applying transformations in textual data is an interesting but totally different question.
>
> To your question on a simple characterization of realizable/agnostic learnability: To clarify,  we have three settings in this paper, which are the invariantly realizable setting, the relaxed realizable setting and the agnostic setting. The relaxed realizable setting and agnostic setting are analogous to the realizable and agnostic setting in classical PAC learning. And we actually have a single simple characterization $VC_{ao}$ for both settings (Theorem 7 and 8). For the invariantly realizable setting, it has a different characterization $VC_o$ because it has a further constraint on the target function, i.e., the target function has to be invariant (see line 121-122 for a formal definition). Hence, we have a different characterization in this setting since we have more prior information about the target function.
>
> To your question on a simple principle as ERM in PAC learning: To clarify, we indeed have a simple principle called ERM-INV (which is a variant of ERM) in the relaxed realizable and agnostic settings. This algorithm is mentioned in line 363, and the details are provided in Appendix H (line 773-780) due to space limitations. In the invariantly realizable setting, as mentioned above, there is more prior information about the target function and because we exploit this, our algorithm for this setting is different. Whether there exists a unifying optimal learning rule in all three settings is an interesting open question.
>
> We didn’t fully understand your last question "if we perform data augmentation under standard PAC learning, but define the sample size as the number of examples in the original (non-augmented) dataset, how does this compare with your estimate of the sample complexity in the invariantly realizable case?". To clarify, throughout the paper and in all of our settings, we define the sample complexity as the number of examples needed from the original distribution (without any augmentations).  If "performing DA under standard PAC learning" means applying DA without transformation invariance assumption, then DA will fail in the worst case since the transformed data might not be labeled the same as the original data (e.g., applying rotations in MNIST). We are happy to clarify more if you could elaborate on this or have more questions.

---

> > ### Comment · Reviewer_vPzm · 2022-08-03
> > **Thank you**
> >
> > Oh right, PAC learning from augmented data may fail without transformation invariance assumption.
> >
> > I am satisfied with your answers to my questions. I will update my score.
> >
> > Thank you for the detailed response.

---

### Official Review · Reviewer_Qyfx · 2022-07-10

**Rating:** 8
**Confidence:** 3
**Soundness:** 4 excellent
**Presentation:** 3 good
**Contribution:** 4 excellent

**Summary:**

### **Brief summary**

This paper studies the sample complexity of PAC learning in a setting where the inputs satisfy invariant properties generated by a group. From a theoretical perspective, this paper compares the sample complexity from such a setting to the sample complexity of ERM, and use this as a lens to theoretically reason about data augmentation, which is a popular heuristic but "is unclear whether and when data augmentation helps."

### **Contribution**

The set up for learning under transformation invariances, as defined by the paper, is as follows: (i) Given a group of transformations, the *augmented data set* is the set of transformed data. (ii) It is assumed that the labels of the data are invariant. Based on this, a hypothesis is invariant to a transformation group if it predicts the same label for a transformed data as the original data.

The results of this paper are shown under three settings according to the level of realizability of the hypothesis class.
- First is the *invariantly realizable PAC learning* setting, which captures scenarios where there exist an invariant hypothesis that makes perfect predictions in the augmented data distribution.
- Second is the *relaxed realizable PAC learning* setting, which captures scenarios where there exist a hypothesis (which may or may not be invariant) that makes perfect predictions in the augmented data distribution.
- Third is the *agnostic* setting, for which there may not exist a perfect predictor for the augmented data distribution.

Then, the contribution of this paper is to give nearly tight upper and lower bounds on the sample complexity of DA--ERM over the augmented data--for all three settings.
- For setting one, the sample complexity of DA is given by the VC dimension across the orbits of the augmented data.
    - However, it is also shown that the optimal sample complexity is instead determined by the VC dimension of the orbits, where the labelings need to be realized by an invariant hypothesis.
- For setting two, the sample complexity of DA depends on a certain graph orientation.
    - On the other hand, the optimal sample complexity for learning this setting is given by the VC dimension across orbits (i.e. without requiring the invariant hypothesis)
- For the agnostic setting, the optimal sample complexity also depends on the VC dimension across orbits.

**Questions:**

- In the problem setup, it seems that the transformations are assumed to be known a prior. What if they are not given? For example, works like Autoaugment often require finding the "right" transformation to apply for a particular task/dataset.

**Limitations:**

The authors provided a discussion of the limitations of this paper in the main text. The potential negative societal impact is not discussed (possibly but due to the technical nature of the paper); this should be addressed in a revised version.

**Strengths And Weaknesses:**

### **Strength**

- The sample complexity bounds concerning DA depend on the VC dimension of/across the orbits, depending on the exact setting. These bounds provide a new theoretical lens to examine data augmentation. For example, it is argued from the VC dimension bounds that
    - In the invariantly realizable setting, DA helps compared to ERM. However, it is not necessarily optimal.
    - In the relaxed realizable setting, DA may actually hurt.
- This paper also proposes an ERM-based algorithm that achieves the optimal sample complexity. The design of this algorithm is quite interesting in my opinion: It works by first checking whether the test instance can be generated by a transformation of the input. If not, it outputs the ERM prediction.
- This paper first provides a rigorous mathematical definition of PAC learning under group transformations. I find this formulation nice to read. The contributions and their proofs are well laid out.
- I find the writing of this paper to be easy to follow. I have some minor comments mentioned below.
- The discussions section provides a candid narrative of the limitations of this paper.

### **Weakness**

- The discussion concerning the 1-inclusion-graph, starting from L273, could be improved to better explain this concept.
- In the relaxed realizable setting, the sample complexity of DA depends on a certain edge orientation notion $\mu(\mathcal{H}, \mathcal{G})$. It might help provide some more explanations to give the intuition behind the notion.
- Some of the proofs in the appendix could use more work to give greater clarity.
    - L762: The sentence following "Every labeling" is broken.

---

> ### Author Response · Authors · 2022-08-02
> **Response to Reviewer Qyfx**
>
> Thanks for your review, especially for checking the details in the appendix!
>
> You are totally correct that the transformations are assumed to be known in this work. Learning under unknown transformations will be an interesting future direction. One intuitive way is applying an augmentation policy searching method like Autoaugment to learn the transformations first and then using the methods in this work based on the learned transformations. But then the results should depend on the quality of the transformations we have learned.

---

### Official Review · Reviewer_vznS · 2022-07-11

**Rating:** 7
**Confidence:** 3
**Soundness:** 3 good
**Presentation:** 3 good
**Contribution:** 3 good

**Summary:**

This paper studies the PAC learnability of hypothesis classes with different levels of invariance realizability, assuming that the data distribution is group-transformation-invariant. For each realizability case, the authors derive sample complexity bounds for data-augmented ERM and a 1-inclusion-graph-based learning algorithm, using two “invariant versions” of VC-dimension that limit the choices of samples and hypotheses. They show that the sample complexity of the former algorithm is sub-optimal or even worse than ERM while the latter one achieves the optimal sample complexity.


**Questions:**

1. I suggest that the authors give an intuitive explanation for the term “distinguish between the original and transformed data” in the main text.
2. Readers could also benefit from an intuitive description about the 1-inclusion-graph-based learning algorithm in the main text.

**Limitations:**

Yes, the authors address the limitation of assuming invariant data distributions.

**Strengths And Weaknesses:**

### Strengths:
1. This paper addresses two important problems: PAC learnability under transformation invariance and theoretical understanding of data augmentation.

2. The theoretical analyses are sound and interesting — the two “invariant versions” of VC-dimension are novel and the lower and upper sample complexity bounds in each realizatiliby case are comprehensive.

3. The conclusions, suggesting that the DA can be sub-optimal in practice and using the 1-inclusion-graph-based algorithm could help achieve the optimal sample complexity, are inspiring.

### Weaknesses:
1. My major concern is to what extent the theoretical results in this paper hold on real data and how much practical insight we could gain from this paper. Specifically,
    1. The two proposed combinatorial complexity measures seem hard to evaluate in practice. Evaluating the VC-dimension of a neural network class while limiting data points to lie in disjoint orbits seems non-trivial to me. I wonder how many previous results on VC-dimension calculation for neural networks could be adapted here. If we cannot evaluate the proposed combinatorial complexity measures in practice, how can we gain insight from the sample complexity results based on it?
   2. Many of the main theoretical results, including the arbitrarily large gap between different versions of VC-dimension and all the lower bounds of the sample complexity (which are used to show the sub-optimality and harm of data augmentation), are based on specific data distribution constructions. I understand that these are common proof practices for VC-dimension but am also wondering to what extent they hold on real data?

2. Some descriptions could be made clearer. For example, throughout the paper I do not see how the probability measure on the transformation group is defined for the data-augmented ERM.

---

> ### Author Response · Authors · 2022-08-02
> **Response to Reviewer vznS**
>
> Thank you for your review!
>
> For evaluating the proposed combinatorial complexity measures, we agree that evaluating these measures will give us a better understanding of the quantitative gaps introduced in this work, and evaluating them efficiently is non-trivial in practice. How to evaluate them in neural networks would be an interesting future direction. Without evaluating these measures, we can still gain some qualitative insights, e.g., DA is *suboptimal* (implied by Theorem 2); DA can *harm* in some cases (implied by Theorem 5). There are also some practical insights for algorithm design. For example, our optimal algorithm in the invariantly realizable setting (in Section 4.2) takes the invariance over the test point into account, which provides some theoretical justification for test-time adaptation such as [Wang et al., 2021].  Our Theorem 5 shows that distinguishing between the original and the transformed data (which means that algorithms are only given the augmented data $GS_{trn}$ without knowing the original training set $S_{trn}$) is important in the relaxed realizable setting and agnostic setting, which provides some high-level hints on designing practical algorithms.
>
> To your question about to what extent the theoretical results hold on real data:
> * The arbitrarily large gaps between the different versions of VC-dimension: To clarify, the gaps between different dimensions do not depend on data distributions as these dimensions are distribution independent. But the gaps depend on instance space X, transformation set G and hypothesis class H and they can be large in real cases. Consider a simple case, where the natural data is k images and the transformation set is rotation by 0, 360/n, 2*360/n, ..., (n-1)*360/n degrees for some integer n. For an expressive hypothesis class (e.g., neural networks) that can shatter all rotated versions of these images, we have VC dimension = nk and vco = vcao = k. For a hypothesis class composed of all hypotheses labeling all upright images and their upside-down variations differently, we have VC dimension = (n-1)k, vcao = k and vco = 0.
> * The lower bounds of sample complexity: The constructions in the lower bounds of the sample complexity depend on data distribution. But the dependence on the dimensions (ignoring the dependence on $\epsilon$) holds for a wide class of data distributions. Let's consider a specific example of classifying land birds vs water birds. The natural data is k images of land birds with land background and water birds with water background. The transformation contains keeping the current background and changing the background from land (water) to water (land).
>   * For the lower bound used to show the sub-optimality of DA (Theorem 2): Let the marginal data distribution be any distribution with support being half of the k original images (which can be extended to any constant fraction of k images). Let the hypothesis class be composed of all hypotheses classifying exactly half of the k images with the original background incorrectly (and the other k/2 images with the original background and all k images with changed background correctly). We can show that DA needs at least vcao = k/2 samples using the same proof techniques of our Theorem 2. But the optimal algorithm only needs zero samples to achieve zero error given that we are in the invariantly realizable setting, which shows the sub-optimality of DA.
>   * For the lower bound used to show the harm of DA (Theorem 5): If we consider a simple hypothesis class, which predicts only based on backgrounds. That is to say, $H=$ {$h_1,h_2$} with $h_1$ predicting all images with water background as water birds and $h_2$ predicting all images with water background as land birds. Let the data distribution be any distribution with distribution support being all the original images. Then given any training data, $h_1$ and $h_2$ have the same empirical loss on the augmented training data. Thus, for any unobserved image, DA will make a mistake with constant probability. Hence DA requires at least $\Omega(k)$ samples. This can be proved by the proof techniques of Theorem 5 with Theorem 6. But it is direct to check that we only need one sample if applying standard ERM, showing the harm of DA.
>
> We will add the real data examples in the revised version. Thanks for your question!
>
> Probability measure on the transformation group: To clarify, we didn't define any probability measure on the transformation group. As introduced in line 160-165, if there exists an $h\in H$ s.t. $h(x) = y$ for all $(x, y) \in G S_{trn}$, data-augmented ERM will output such an $h$; If there does not exist such an $h$, data-augmented ERM outputs an $h\in H$ with the minimal loss $L(GS_{trn}, h)$. Here the loss can be any function, which can be defined based on any probability measure on the transformation group. Thanks for pointing this out and we will make it clearer.
>
> We will clarify the 1-inclusion-graph algorithms further.

---

> > ### Comment · Reviewer_vznS · 2022-08-09
> > **Thanks for your response**
> >
> > My concerns are addressed. I have increased my score by one. Looking forward to seeing the revised version.

---

### Meta-Review · Area_Chair_Jauh · 2022-08-31

**Recommendation:** Accept
**Confidence:** Certain

**Metareview:**

All reviewers are positive.

Clear accept.

**Award:**

Yes

---

### Decision · Program_Chairs · 2022-09-14

Accept